# Promoting photocatalytic $CO_2$ reduction with a molecular copper purpurin chromophore

Huiqing Yuan[1], Banggui Cheng[1], Jingxiang Lei[1], Long Jiang [1] & Zhiji Han [1✉]

$CO_2$ reduction through artificial photosynthesis represents a prominent strategy toward the conversion of solar energy into fuels or useful chemical feedstocks. In such configuration, designing highly efficient chromophores comprising earth-abundant elements is essential for both light harvesting and electron transfer. Herein, we report that a copper purpurin complex bearing an additional redox-active center in natural organic chromophores is capable to shift the reduction potential 540 mV more negative than its organic dye component. When this copper photosensitizer is employed with an iron porphyrin as the catalyst and 1,3-dimethyl-2-phenyl-2,3-dihydro-1H-benzo[d]imidazole as the sacrificial reductant, the system achieves over 16100 turnover number of CO from $CO_2$ with a 95% selectivity (CO vs $H_2$) under visible-light irradiation, which is among the highest reported for a homogeneous noble metal-free system. This work may open up an effective approach for the rational design of highly efficient chromophores in artificial photosynthesis.

[1] MOE Key Laboratory of Bioinorganic and Synthetic Chemistry, School of Chemistry, Sun Yat-sen University, Guangzhou 510275, China.
✉email: hanzhiji@mail.sysu.edu.cn

Artificial photosynthesis (AP) represents an appealing strategy for the conversion of solar energy into stored chemical potential energy[1,2]. Inspired by natural photosynthesis, photocatalytic reduction of $CO_2$ to value-added chemicals (such as CO, $CH_3OH$, and $C_{\geq 2}$ products) is a viable solution to obtain sustainable energy in closing the anthropogenic carbon cycle[3–7]. Several approaches are available for photochemical reduction of $CO_2$ to access fuels or chemical precursors[8]. Homogeneous AP systems usually offer distinct advantages with respect to structural optimization and product selectivity[9]. For example, the reduction potentials of all components can be finely tuned through functional group manipulation to facilitate efficient light-driven electron transfer[10]. In a typical homogeneous AP system, a photosensitizer (PS) is a key component responsible for light-harvesting and photochemical reactions[10,11]. To achieve a high activity in photocatalytic $CO_2$ conversion, it is essential to develop satisfactory photosensitizers that have large absorption cross sections in the visible region and are capable of promoting efficient multiple electron transfer processes. Recent studies on precious metal-based PS (such as Ru, Ir, and Re) have shown good activity for $CO_2$ reduction[12–25]. However, molecular chromophores comprising earth-abundant elements usually give relatively low turnover numbers (TONs), which is a significant limitation in developing inexpensive AP systems[26–29]. Thus, developing highly efficient noble-metal-free PS for $CO_2$ conversion remains a persistent challenge for the scientific community.

Anthracene organic dyes, which is an emerging group of inexpensive PS with high molar absorption coefficients in the visible region, have demonstrated to have good stability in photocatalytic $CO_2$ reduction[30]. However, the activity of these photocatalytic systems is much lower than the ones using metal-based photosensitizers, due to the inefficiency in multiple electron transfer to the catalyst[11,31]. The system containing 9-cyanoanthracene as the PS and an iron(0) tetrapenylporphyrin as the catalyst produces CO with a turnover frequency (TOF) of ~1.5 h$^{-1}$[32]. Since 2016, purpurin (PP) (the main coloring matter in natural madder) has been studied as one of the most active organic photosensitizers for $CO_2$ reduction when coupled with a variety of metal catalysts[33–35], achieving $TON_{CO}$ as high as 1365 (TOF ~ 300 h$^{-1}$) with respect to the catalyst[33]. Recently, because of the highly luminescent and tunable redox properties, copper diamine photosensitizers have shown promising activities in driving the reduction of $CO_2$[12,36]. Ishitani et al. have shown that dimeric Cu complexes with heteroleptic diimine and phosphine ligands are effective photosensitizers to exhibit 273 turnovers of CO in 12 h[37]. The systems employed various copper diimine diphosphine complexes as the photosensitizers, Fe[37–39], Co[40], Ni[41], or Mn[42,43] metal complexes as the catalysts, and 1,3-dimethyl-2-phenyl-2,3-dihydro-1H-benzo[d] imidazole (BIH) as the sacrificial electron donor that produced CO up to 2680 turnovers under optimal conditions[40]. While Cu diimine photosensitizers exhibit relatively higher activities in photocatalytic $CO_2$ reduction than organic PS, they require much higher energetic light irradiation owing to their absorption spectra centered at the UV region[36]. To facilitate fast electron transfer between an electron donor and acceptor, various PS-catalyst dyads have been studied[8,44,45]. This strategy has shown an increase in activities from the linked systems, with $TON_{CO}$ as high as 4347[46]. However, these studies were performed with precious metals, and fastback electron transfer from the catalyst to the PS is usually observed in these dyads, which could hinder efficient charge separation.

In this work, we report a highly active copper purpurin PS for visible-light-driven $CO_2$ reduction to CO in a noble metal-free system (Fig. 1). By coordinating a redox-active Cu center directly to the redox noninnocent catecholate motifs[47] on natural organic dyes, we create a simple yet highly efficient PS to facilitate

challenging multielectron reductions. Electrochemical studies reveal that the incorporation of Cu in purpurin produces a much more reducing PS for catalysis. When coupled with a Fe porphyrin as the catalyst and BIH as the sacrificial electron donor, this Cu PS significantly boosts the photocatalytic activity (>50-fold) of the system for $CO_2$ reduction compared with its organic dye component. This homogeneous system achieves over 16,100 turnovers of CO and a $TOF_{max}$ of 7650 per hour with a 95% selectivity (CO vs H$_2$), which is the highest for homogeneous noble metal-free systems. Importantly, this system demonstrates a >fourfold increase in $TON_{CO}$ compared to the most active noble metal-based systems using metal porphyrin catalysts[48].

## Results and discussion

**Preparation and characterization of CuPP.** As outlined in Fig. 2, Cu purpurin complex was prepared by heating $Cu(NO_3)_2$, PP (2.0 equiv), and $NaHCO_3$ (4.0 equiv) in dimethylformamide (DMF) at 100 °C for 24 h. X-ray diffraction (XRD, *vide infra*) quality single crystals of the isolated compound have not been obtained due to its low solubility in most organic solvents. Electrospray ionization mass spectrometry (ESI-MS) and elemental analysis of this product suggest the formation of a $Na_2Cu(PP)_2$ complex (Supplementary Fig. 1a). Cation exchange of this compound was carried out in water with tetrabutylammonium bromide (TBABr) (2.0 equiv). Recrystallization of the resulting $(TBA)_2Cu(PP)_2$ (CuPP) by slow diffusion of diethyl ether into an acetonitrile solution gave dark-red crystals suitable for single-crystal XRD studies.

Polyhydroxy-9,10-anthracenediones have been used for the detection of various transition metal ions[49,50]; however, solid-state structures of these complexes are very rare[51,52]. Figure 2 shows the crystal structure of a Cu purpurin complex. In this structure, the Cu center displays a square planar geometry and is coordinated with the 1- and 2-hydroxyl groups on the 9,10-anthracenediones, which is different from the 2,3-oxygen chelates proposed in the literature based on theoretical calculation[53]. The Cu–oxygen distances (Cu–O1 = 1.927(2) and Cu–O2 = 1.917(2) Å) are similar to the Cu–O bonds reported for Cu(II) catecholate compounds in the literature (Supplementary Table 1). This evidence along with the electronic charge balancing, and paramagnetic proton resonances in the $^1$H NMR spectra (Supplementary Figs. 3 and 4) are all consistent with the assignment of a Cu(II) center in CuPP. The catecholate carbon–oxygen distances (C2–O2 = 1.295(3) and C1–O1 = 1.303(3) Å) are identical to those found for a Ru $(CO)_2(PBu_3)_2(1,2,3$-trihydroxy-9,10-anthracenedione) complex, suggesting they are single C–O bonds[51]. In the anthraquinone moieties, the C4–O3 distance (1.239(3) Å) is typical a double bond, while the other double bond C11–O4 (1.274(3) Å) is elongated due to the hydrogen bonding interaction with its adjacent hydroxyl.

The UV–vis absorption spectra of CuPP and PP were measured at room temperature in DMF (Fig. 3a). CuPP displays a much stronger visible absorption profile than PP, with intense absorption that tails to almost 650 nm (Fig. 3a). The significant red shift of the maximum absorption band at 478 nm ($\varepsilon = 9.94 \times 10^3$ M$^{-1}$ cm$^{-1}$) of PP to 566 nm ($\varepsilon = 3.85 \times 10^4$ M$^{-1}$ cm$^{-1}$) of CuPP is probably due to the generation of dianion of hydroxyanthraquinone and an induction effect from the Cu(II) center. Linear Beer's law at two wavelengths (278 and 566 nm) applied for the range of concentrations used in CuPP (Supplementary Fig. 10), suggesting negligible solute–solute interactions.

Because the UV–vis spectrum of CuPP exhibits identical features as compared to the dianion form of PP in the visible region[54], the absorption band at 566 nm is assigned to the charge transfer band on the PP ligands. The incorporation of Cu(II)

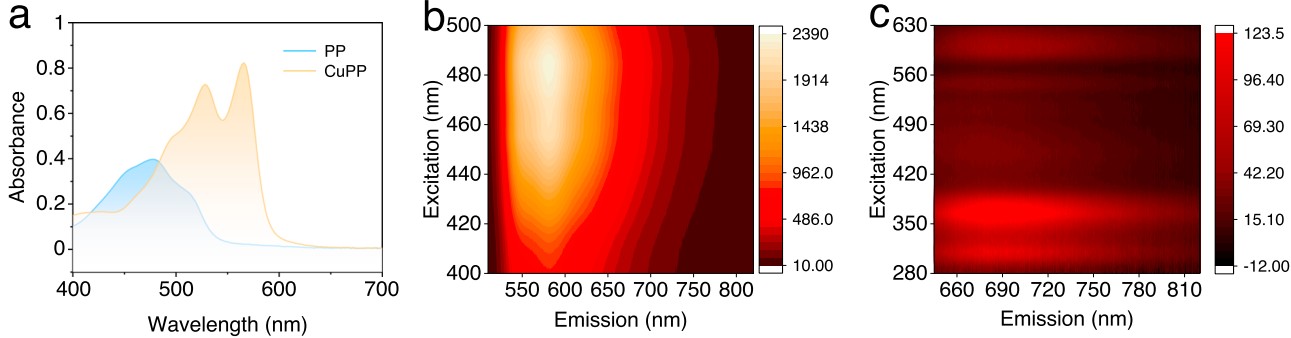

**Fig. 1 Illustrative scheme of the photocatalytic CO₂ reduction system.** BIH is the sacrificial reductant, and CuPP is the photosensitizer, and FeTDHPP is the catalyst investigated in this study.

**Fig. 2 Synthetic route for Cu purpurin complexes and crystal structure of CuPP.** Thermal ellipsoids are shown at the 50% probability level. Parts of hydrogen atoms and TBA cations are not shown for clarity.

**Fig. 3 Absorption and emission profiles of CuPP and PP. a** UV–vis analysis of CuPP (20 μM, orange) and purpurin (40 μM, blue) in DMF; excitation and emission spectra of PP (**b**) and CuPP (**c**) at different wavelengths at 298 K in DMF. Source data are provided as a Source Data file.

enhances this charge transfer process by showing a much higher molar extinction coefficient (Supplementary Fig. 10). The photoluminescence quantum yield for CuPP ($8.2 \times 10^{-3}$) is lower than that of PP ($2.7 \times 10^{-2}$) (Supplementary Table 2). Instead, Grazia et al. observed an increase of the fluorescence intensity when adding Al(III) to PP[55]. Short stokes shifts of emissions were observed for both Al-PP and PP, and their excitation spectra are comparable with the absorption spectra (Supplementary Fig. 11a)[55,56], suggesting that these compounds undergo relatively low structural reorganization between the ground and excited states. In contrast, CuPP displays structureless emission at 693 nm with a large bathochromic shift. The distinct absorption and emission spectra of CuPP (Supplementary Fig. 12b) indicate a significant structural change in its excited state. Since Cu(I) complexes tend to adopt distorted geometries compared to their Cu(II) analogs[40,57–60], the nature of the emission of CuPP is likely from a Cu(I) excited state generated from a LMCT process. Excitation–emission spectra (Fig. 3b, c) show that the emission of CuPP is most intense when excited at 375 nm. The excited-state

lifetimes are on the nanosecond timescale for both CuPP ($\tau = 0.98$ ns) and PP ($\tau = 1.1$ ns)[35,54].

**Electrochemical analysis of CuPP.** Figure 4 shows the cyclic voltammograms (CVs) of CuPP and PP in DMF solution containing 0.1 M tetrabutylammonium hexafluorophosphate (TBAPF₆). Indeed, the incorporation of a Cu(II) center into PP creates additional reduction events. CuPP displays three irreversible reduction potentials at −1.05, −1.50, and −1.69 V vs SCE. The most negative redox couple of CuPP at −1.75 V is 540 mV more negative than the PP²⁻/PP⁻ couple (Fig. 4a)[33]. Square wave voltammetry (SWV) reveals that there are four successive reductions, which correspond to 2, 1, 1, and 2 electron processes (Supplementary Fig. 17).

To explore the electron requirements for photocatalytic systems, further CV experiments were performed to study the electrocatalytic activity for CO₂ reduction. Under an atmosphere of CO₂, CuPP shows a quasi-reversible reduction wave at −0.97 V and a catalytic wave that starts to appear below −1.89 V as

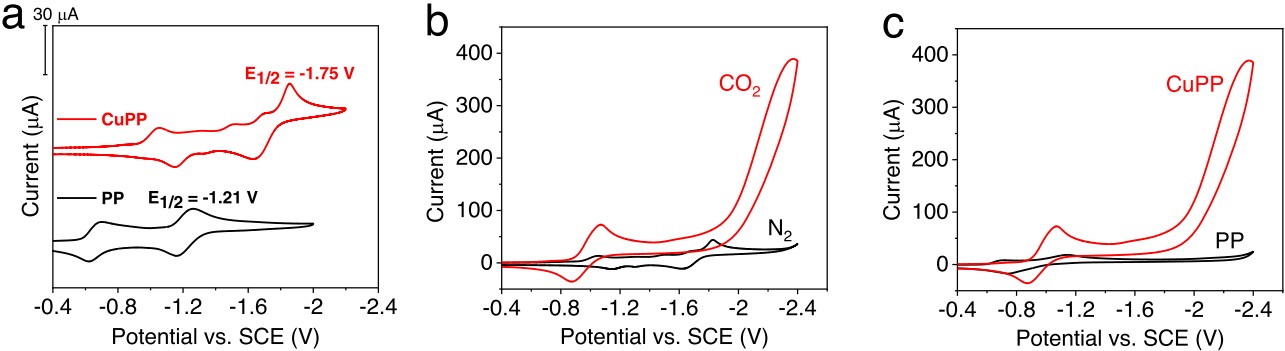

**Fig. 4 Electrochemical data of CuPP and purpurin under $N_2$ or $CO_2$.** Experiments were performed in DMF containing 0.1 M TBAPF$_6$ at scan rate 0.1 V/s: **a** 1 mM CuPP (red) and 1 mM PP (black) under $N_2$; **b** 1 mM CuPP under $N_2$ (black) and $CO_2$ (red); **c** 1 mM CuPP (red) and 1 mM PP (black) under $CO_2$. Source data are provided as a Source Data file.

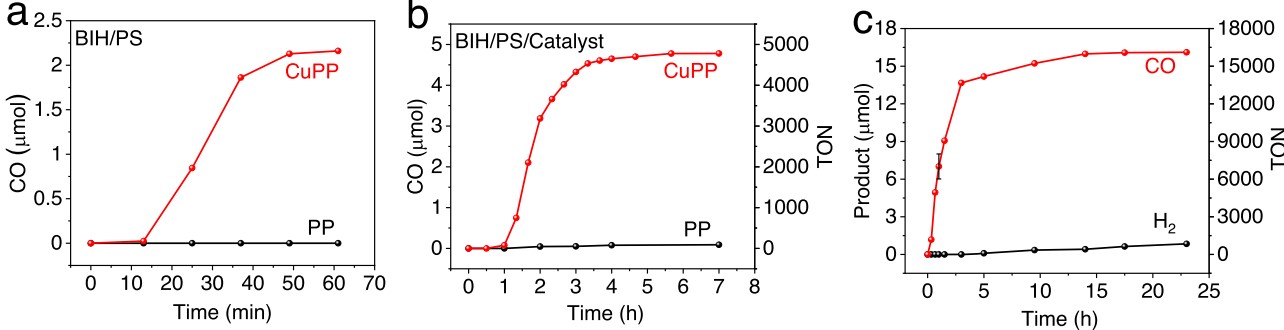

**Fig. 5 Photocatalytic $CO_2$ reduction.** Experiments were performed in $CO_2$-saturated DMF solution containing **a** 0.1 mM CuPP (red) or 0.2 mM PP (black), and 100 mM BIH; **b** 0.2 mM PP (black) or 0.1 mM CuPP (red), 0.2 μM FeTDHPP, and 10 mM BIH; **c** 0.1 mM CuPP, 0.2 μM FeTDHPP, and 100 mM BIH. Error bar (95% confidence interval based on three runs) is included at the 1-h point, which is the typical error in the measurement (14%). Source data are provided as a Source Data file.

compared to no observation of current enhancement in $N_2$ (Fig. 4b), suggesting that CuPP could act as a $CO_2$ reduction catalyst. CV control experiments for PP and electrolyte solution show little current enhancement at the same potentials in the presence of $CO_2$ (Fig. 4c). The integrals of SWV show a total of four-electron reduction at −1.13 V prior to the catalytic current (Supplementary Fig. 17). The different electrochemical profiles observed for CuPP in the presence of $CO_2$ and $N_2$ indicate that the four quinone moieties on the PP ligands undergo four-electron reductions and four complexations of $CO_2$ to generate carbonates. In fact, a similar observation for 9,10-anthracene-diones has been reported in the literature[61]. Controlled potential coulometry experiments, performed at −1.89 V (vs SCE) in $CO_2$-saturated DMF, show that CO is the main gas product with a faradaic yield of 17% and only a negligible amount of $H_2$ generated (Supplementary Fig. 20). In a control experiment without CuPP, there is little CO or $H_2$ produced. These data confirm that the observed current enhancement of CuPP corresponds to $CO_2$ reduction, not proton reduction.

**Photocatalytic $CO_2$ reduction.** The electrocatalytic $CO_2$ reduction properties of CuPP (see Fig. 4) together with its strong absorption in the visible spectral region (Fig. 3) provide strong evidence that CuPP may serve as both photosensitizer and catalyst for $CO_2$ reduction. Photocatalytic experiments were performed by irradiating CuPP (0.1 mM) in DMF with a white light-emitting diode ($\lambda > 400$ nm) in the presence of BIH (0.1 M) as the sacrificial donor. The amount of gaseous products were monitored in real time using gas chromatography (GC). Figure 5a shows that CuPP-containing system generates CO upon

irradiation with an initial turnover frequency of 4.3 h$^{-1}$, while there is no CO observed in the control system using PP (2.0 equiv) instead of CuPP (Table 1). However, the rate of CO evolution decreases dramatically after one hour. The disappearance of the color of the reaction mixture after irradiation indicates the decomposition of CuPP.

The activity and stability of photocatalytic $CO_2$ reduction can be significantly improved with the addition of chloro iron (III) 5, 10, 15, 20-tetrakis (2',6'-dihydroxyphenyl)-porphyrin (FeTDHPP) as the cocatalyst (Fig. 1), which has been shown to be an active electrocatalyst for $CO_2$ reduction[62]. In a typical experiment, the system containing 0.1 mM CuPP, 0.2 μM FeTDHPP, and 10 mM BIH, produces 4779 TONs of CO/mol of catalyst in 7 h in $CO_2$-saturated DMF solution (Fig. 5b and Table 1). For comparison, an experiment performed using PP as the PS under the same conditions gives a TON of 88 for $CO_2$ (Fig. 5b and Table 1), underscoring the importance of CuPP for photocatalytic $CO_2$ reduction. The CuPP PS can also be generated in situ with the addition of 1:2 ratio of Cu$^{2+}$ and PP. The amount of CO produced with the in situ-generated PS is ~25% lower than that of the system using isolated CuPP (Supplementary Fig. 21a). UV–vis spectra confirm that CuPP can be generated in minutes when mixing Cu$^{2+}$ and PP in the presence of BIH (Supplementary Fig. 21b). Control experiments show that the absence of PS leads to no CO or $H_2$ production (Supplementary Table 4).

An induction period observed at low BIH concentration disappears as [BIH] increased above 0.1 M (Supplementary Fig. 22), which suggests that the rate of CO production is limited by the concentration of sacrificial donor. At 100 mM concentration of BIH, the activity of $CO_2$ to CO conversion is further improved, achieving 16,109 TONs in 23 h with an initial TOF of

**Table 1 Control and other photocatalytic CO₂-reduction experiments.**

| Entry | PS | [FeTDHPP] (µM) | [BIH] (mM) | Irradiation time (h) | CO (µmol) | H₂ (µmol) | TON$_{CO}$ | TOF$_{CO}$$^{max}$ (h⁻¹) | TON$_{H2}$ | Sel$_{CO}$ (%) |
|---|---|---|---|---|---|---|---|---|---|---|
| 1 | PP | 0.2 | 10 | 7 | 0.088 | 0 | 88 | 45 | 0 | 100 |
| 2 | CuPP | 0.2 | 10 | 7 | 4.779 | 0.27 | 4779 | 1356 | 270 | 95 |
| 3 | CuPP | 0.2 | 100 | 23 | 16.109 | 0.843 | 16109 | 7650 | 843 | 95 |
| 4 | CuPP | 0 | 100 | 1 | 2.2 | 0 | 4.4a | 4.3 | 0 | 100 |

aCalculated by the equation TON$_{CO}$ = n(CO)/n(CuPP).
Comparison data for the photocatalytic systems in CO₂-saturated DMF solution.

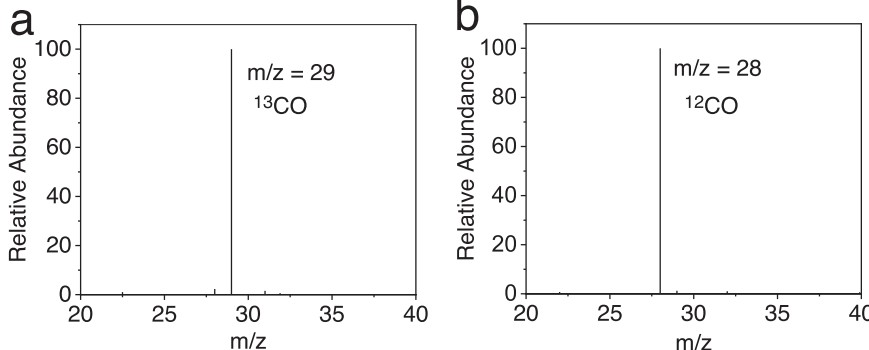

**Fig. 6 GC/MS chromatograms of carbon monoxide.** Photocatalytic CO₂ reduction in ¹³CO₂-saturated (**a**) and ¹²CO₂-saturated (**b**) DMF solutions containing 1.0 µM FeTDHPP, 100 mM BIH, and 0.1 mM CuPP.

7650 h⁻¹ with respect to [FeTDHPP] (Fig. 5c and Table 1), which is more than ten times more active than previously reported systems using PP as the PS (Supplementary Table 6). The activity of the photochemical systems was optimized further by varying the [CuPP] and [FeTDHPP]. At fixed concentrations of BIH (100 mM) and FeTDHPP (2 µM), increasing the CuPP concentration increases the overall rate of CO production (Supplementary Fig. 23). However, the rate of CO generation does not increase above 0.1 mM concentration of CuPP. When the CuPP concentration is fixed at 0.1 mM and [FeTDHPP] is varied (Supplementary Fig. 24), the catalyst is most active (on a TON basis) at low concentrations. The total amount of CO evolved increases up to 40 µM FeTDHPP, suggesting that the activity of the system becomes limited by electron transfer to the catalyst. For a system containing 2.0 µM FeTDHPP/100 mM BIH/0.1 mM CuPP, the initial 1 h quantum yield of CO₂ to CO conversion at 450 nm was determined at 6.0 ± 0.6%.

Quantification of the gaseous products shows an overall selectivity of 95% for CO and 5% for H₂. In the solution, 1.4 µmol formic acid was quantified by high-performance liquid chromatography (HPLC). However, a similar amount (1.5 µmol) of formic acid was detected from the experiment performed under N₂, suggesting that formic acid is not generated from CO₂ reduction. In fact, formic acid generated from hydrolysis of DMF has been previously reported[63]. To confirm that CO₂ is the source of carbon in photolysis, isotopic labeling experiments were performed under an atmosphere of ¹³CO₂. The GC–MS analysis of the gaseous products shows a diagnostic peak of ¹³CO (Fig. 6). It reveals that CO is derived from CO₂ reduction rather than decomposition of other components in the photocatalytic system.

To evaluate the homogeneity of the system, excess of Hg (0.02 mL, 2700 equiv of CuPP) was added to the reaction vessel at the beginning of the experiments, the activity of CO generation remains identical over the course of photolysis (Supplementary Fig. 25). In addition, dynamic light-scattering (DLS) experiments show there is no detectable nanoparticle generated in the reaction mixture before and after CO₂ reduction (Supplementary Fig. 26).

These control experiments suggest that there are no Cu colloids generated and the CuPP/FeTDHPP/BIH system stays homogeneous during photocatalytic CO₂ reduction.

Several additional experiments were carried out to probe the mechanism of decomposition of the system. When the rate of CO production decreases significantly, fresh solutions containing the same amounts of CuPP, FeTDHPP, BIH, or a mixture of them were added to the reaction mixtures at 23 h. Addition of one or two components leads to much less recovery of the original activity, compared to the one that added all three components (Supplementary Fig. 28). This suggests that the decomposition of all three components occurs during CO generation. Furthermore, absorption of the reaction mixture containing CuPP, FeTDHPP, and BIH in DMF was monitored by UV/vis spectroscopy during photolysis (Supplementary Fig. 29). The intensity of the absorption band at 400 nm, which corresponds to the reductions of PP ligands[35], decreases gradually on the course of irradiation. This confirms that CuPP decomposes during CO₂ reduction.

**Mechanistic studies.** Further experiments were performed to study the photochemical mechanism of CO₂ reduction. UV/vis studies show there is no change of the CuPP spectrum with the addition of BIH or FeTDHPP under CO₂ (Supplementary Fig. 30), suggesting that the first electron transfer step is initiated by photon absorption. Photochemical steps were studied through fluorescence quenching of CuPP in DMF. The reductive quenching rate constant (using BIH as the quencher) is 9.23 × 10⁹ M⁻¹ s⁻¹ based on Stern–Volmer equation (Supplementary Fig. 31). A linear plot for the oxidative quenching could not be obtained due to the fact that FeTDHPP has a nonnegligible absorption at both the excitation and emission wavelengths (Supplementary Figs. 32 and 33), which is in agreement with the literature[32,34]. Because [BIH] used in most of our photocatalytic experiments is more than 5 orders of magnitude higher than [FeTDHPP] and the $k_q$ for reductive quenching is at the diffusion-controlled limit, reduction of CuPP* by BIH is expected to be the first electron-transfer step.

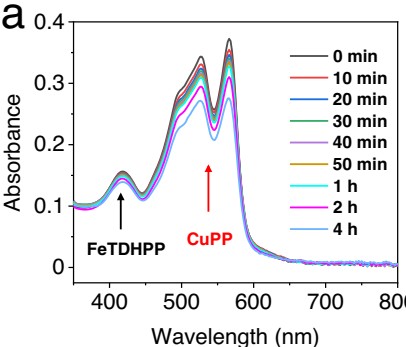
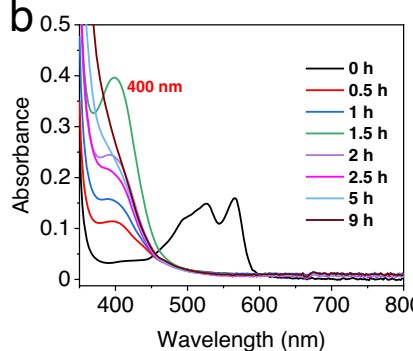

**Fig. 7 UV–vis absorption spectra of systems. a** In all, 0.01 mM CuPP, 0.5 µM FeTDHPP, 0.1 mM BIH in a quartz cuvette (10-mm path length), and **b** 0.1 mM CuPP, 0.2 µM FeTDHPP, and 100 mM BIH upon irradiation with white LED light. Solutions for (**b**) at different times were transferred and diluted five times with DMF to a quartz cuvette (2-mm path length) under $N_2$.

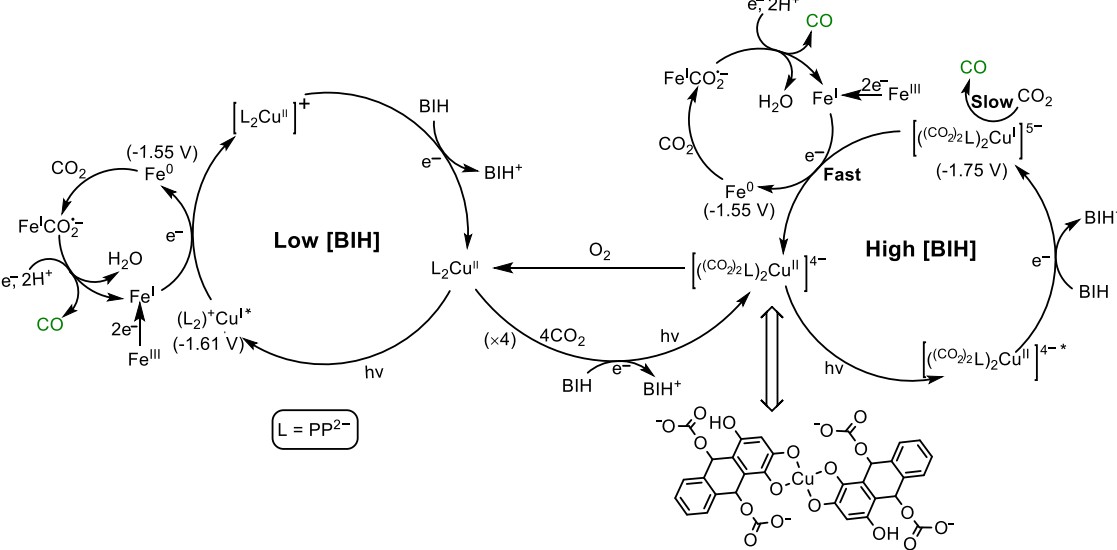

**Fig. 8 Proposed reaction scheme.** Different pathways for photocatalytic $CO_2$ reduction are proposed for low [BIH] (left) and high (BIH) (right).

CO generation was observed even at [BIH] as low as 0.1 mM. UV–vis spectra show no generation of the additional absorption band in the visible region during $CO_2$ reduction (Fig. 7a), indicating CuPP is the active PS at this condition. Since $[L_2Cu^{II}]^-$ (L = $PP^{2-}$) has a much higher reduction potential (−1.05 V) than that of Fe(I)TDHPP ($E_{1/2}$ = −1.55 V), $CO_2$ reduction has to proceed through oxidative quenching of $(L_2)^+Cu^{I*}$ (−1.61 V) (Supplementary Table 3) in the photochemical step (Fig. 8). At high [BIH] (100 mM), a yellow species with an absorption band at 400 nm is generated under visible-light irradiation (Fig. 7b). When the reaction mixture is exposed to the air, the solution returns to purple color and over 80% of the original CuPP is recovered based on UV–vis spectra (Supplementary Fig. 34). Electrochemical studies in the presence of $CO_2$ (Fig. 4b) suggest that CuPP could undergo four steps of reduction and complexation of $CO_2$ to generate a $[(^{(CO2)2}L)_2Cu^{II}]^{4-}$ species (Fig. 8). In fact, PP has been shown to proceed a two-electron reduction to generate a species at 398 nm during photocatalytic $CO_2$ reduction[34]. Thus, we conclude that the intermediate with maximum absorption at 400 nm is a $[(^{(CO2)2}L)_2Cu^{II}]^{4-}$ species. $[(^{(CO2)2}L)_2Cu^{I}]^{5-}$ (−1.75 V), generated by further reduction of $[(^{(CO2)2}L)_2Cu^{II}]^{4-}$ through reductive quenching, is corresponding to electron transfer to the Fe catalyst (Fig. 8).

On the basis of a much faster turnover of $CO_2$ to CO with CuPP compared to PP, electron transfer from the reduced PS to

the catalyst is proposed to be a turnover-limiting step (TOLS) (Fig. 8). Although CuPP is active for photocatalytic $CO_2$ reduction without an additional catalyst, the significantly lower TOF observed in the CuPP/BIH system clearly shows that this process is much slower than the TOLS of the CuPP/FeTDHPP/ BIH system (Fig. 8). Electrochemical studies show that CuPP exhibits a more negative reduction potential (−1.75 V vs SCE) than that of PP (−1.21 V vs SCE). In order to access a previously reported Fe(0) porphyrin intermediate for $CO_2$ reduction (Fig. 8)[62], CuPP is expected to have a larger driving force for reduction and hence a faster rate for electron transfer to the catalyst than PP.

This work shows an example where an active photosensitizer for $CO_2$ reduction has been constructed through direct coordination of a redox-active metal center to natural organic dyes. This Cu photosensitizer displays strong visible-light-harvesting and multiple electrochemical properties. In the AP systems using metal porphyrins as the catalysts, CuPP exhibits great photocatalytic activity for $CO_2$ reduction when compared with previously reported homogeneous photosensitizers, even including the noble metal ones (Supplementary Tables 5–8). For example, the CuPP/FeTDHPP system achieves over 16,100 turnovers of CO with a maximum TOF of 7650 h⁻¹, which is two orders of magnitude more than a reported Ir(ppy)₃/ FeTDHPP system (TON = 140 in 55 h)[32]. This work not only

paves the way to improve $CO_2$ to CO conversion in homogeneous light-driven systems, but also provides perspectives for the rational design of efficient but low-cost photosensitizers for solar fuel production.

## Methods

**Materials.** All solvents and chemicals were commercially purchased and used as obtained without further purification, except for purpurin. Purpurin was recrystallized twice from absolute ethanol before usage.

**Preparation of BIH.** BIH was synthesized from a modified method based on the literature[64]. In total, 6.0 g of 2-phenylbenzimidazole was treated with 16 g of methyl iodide in 30 mL of methanol containing 1.28 g of NaOH. The reaction mixture was heated at 110 °C for 24 h to give a faint yellow solid ($BIH^+I^-$). The crude product was washed with $H_2O$ to decolorize. The yield is over 80%. Then, a solution of $BIH^+I^-$ in methanol was reduced by $NaBH_4$ (2 eq.) under $N_2$ to give a white solid (BIH). The white solid was purified by washing with $EtOH/H_2O$ (5:1, v/v). The yield is over 90%. The synthetic intermediates and target complexes were evidenced by $^1H$ NMR.

**Preparation of chloro iron (III) 5,10,15,20-tetrakis (2′,6′-dihydroxyphenyl)-porphyrin.** Chloro iron (III) 5,10,15,20-tetrakis (2′,6′-dihydroxyphenyl)-porphyrin was prepared following a reported procedure[62]. A solution of 2′–6′-dimethoxybenzaldehyde (1 g, 6.02 mmol) and pyrrole (0.419 mL, 602 mmol) in chloroform (600 mL) was degassed by argon for 20 min, then $BF_3·OEt_2$ (0.228 mL, 0.87 mmol) was added via a syringe. The solution was stirred for under an inert atmosphere in the dark for 1.5 h, and 2,3-dichloro-5,6-dicyano-1,4-benzoquinone (DDQ) (1.02 g, 4.51 mmol) was added to the reaction. The mixture was stirred for an additional 1.5 h at reflux, cooled to room temperature, and 1 mL of triethylamine was added to neutralize the excessive acid. Then the solvent was removed, and the resulting black solid was purified by column chromatography (silica gel, dichloromethane) affording 5, 10, 15, 20-tetrakis(2′,6′-dimethoxyphenyl)-21H,23H-porphyrin as a purple powder (290 mg, 23%). To a solution of 5, 10, 15, 20-tetrakis(2′,6′-dimethoxyphenyl)-21H,23H-porphyrin (200 mg, 0.235 mmol) in dry dichloromethane (10 mL) at 0 °C was added $BBr_3$ (1000 μL, 10.38 mmol). The resulting green solution was stirred for 24 h at room temperature, then placed in ice water. Ethyl acetate was added to the suspension, and the mixture was washed with $NaHCO_3$. The organic layer was separated, washed twice with water, and then dried over anhydrous $Na_2SO_4$. The resulting solution was evaporated. The residue was purified by column chromatography (silica gel, 2:1 ethyl acetate/dichloromethane) to yield 5, 10, 15, 20-tetrakis(2′,6′-dihydroxyphenyl)-21H,23H-porphyrin as a purple powder (150 mg, 87%). Finally, a solution of 5, 10, 15, 20-tetrakis(2′,6′-dihydroxyphenyl)-21H,23H-porphyrin (100 mg, 0.135 mmol), $FeCl_2·4H_2O$ (270 mg, 1.35 mmol), and 2,6-lutidine (39 μL, 0.335 mmol) was heated at 50 °C and stirred for 3 h under an inert atmosphere in dry methanol. After methanol was removed, the resulting solid was dissolved in ethyl acetate, washed with 1.2 M HCl solution until pH was neutral. The crude product was purified by column chromatography (silica gel, ethyl acetate) to give chloro iron (III) 5,10,15,20-tetrakis (2′,6′-dihydroxyphenyl)-porphyrin as a brown solid (100 mg, 89%). The synthetic intermediates and target complexes were evidenced by $^1H$ NMR and HRMS spectroscopy.

**Preparation of CuPP.** A solution of $Cu(NO_3)_2·3H_2O$ (672 mg, 2.7 mmol) in DMF (5 mL) was added to a solution of purpurin (1380 mg, 5.4 mmol) in DMF (50 mL). The orange solution quickly turned into a brown suspension upon addition. Then a solution of $NaHCO_3$ (907 mg, 10.8 mmol) in $H_2O$ (10 mL) was added to the reaction mixture with stirring for 24 h at 100 °C. After the mixture cooled down to room temperature, insoluble solids were removed by suction filtration. The filtrate was concentrated by rotary evaporation to ~10 mL, then excess diethyl ether was added, and the mixture was allowed to stir overnight at room temperature to precipitate out the product. A black precipitate ($Na_2Cu(PP)_2$) was collected under suction filtration, washed with diethyl ether (3 × 50 mL), and dried under vacuum (1.0 g, 60% yield). Anal. Calcd. for $C_{28}H_{12}CuO_{12}Na_2 · 2H_2O$: C, 51.43; H, 2.47; found: C, 51.26; H, 2.47. Cation exchange was carried out with the addition of tetrabutylammonium bromide (TBABr) (522 mg, 1.62 mmol, 2.0 equiv) to $Na_2Cu$ $(PP)_2$ (500 mg, 0.81 mmol) in water at 60 °C for 6 h. The mixture was filtered and a brown solid was washed with water (3 × 10 mL) and then collected. Recrystallization of this compound from vapor diffusion of diethyl ether into an acetonitrile solution gave red crystals amenable to X-ray diffraction studies. The overall yield was 22%. $^1H$ MMR (400 MHz, $d_6$-DMSO, 25 °C): δ = 17.21, 8.18, 7.70, 7.54, 7.18, 6.64, 3.20–3.13, 1.57, 1.32–1.23, 0.93 ppm. All resonances are broad. Anal. Calcd. for $C_{60}H_{84}N_2O_{10}Cu$: C, 68.19; H, 8.01; N, 2.65; found: C, 67.94; H, 8.16; N, 2.60.

**Characterization.** $^1H$ and $^{13}C$ NMR spectra were recorded at a Bruker advance III 400-MHz NMR instrument. HRMS spectra were obtained on a Thermo Scientific Orbitrap Q Exactive ion trap mass spectrometer and QTOF-MS (Bruker Daltonics, times TOF). UV–vis spectra were determined on a Thermo Scientific GENESYS 50 UV–visible spectrophotometer. Elemental analyses were performed at an

Elementar Vario EL analyzer. Dynamic light-scattering experiments were performed with a Brookhaven Elite Sizer zata-potential and a particle-size analyzer. Photoluminescence quantum yields were determined using a FLS 980 (Edinburgh instruments) absolute photoluminescence quantum yield measurement system and integrating sphere as a sample chamber. The steady-state fluorescence of solid samples was measured with FLS 920 fluorescence spectrometer (Edinburgh instruments).

**X-ray crystallography.** X-ray diffraction data were collected on SuperNova single-crystal diffractometer using the CuKα (1.54184 nm) radiation at 100 K. Absorption correction was carried out by a multiscan method. The crystal structure was solved by direct methods with SHELXT[65] program, and was refined by full-matrix least-square methods with SHELXL[65] program contained in the Olex2 suite[66]. Weighted R factor ($Rw$) and the goodness of fit $S$ were based on $F_2$, conventional $R$ factor($R$) was based on $F$ (Supplementary Table 9). Hydrogen atoms were placed with the AFIX instructions and were refined using a riding mode. Figures were drawn with Diamond software. Details can be obtained from the Cambridge Crystallographic Data Centre at www.ccdc.cam.ac.uk for CCDC accession number 2017326.

**Cyclic voltammetry.** Cyclic voltammetry (CV) measurements were performed with a CHI-760E electrochemical analyzer using a one-compartment cell with a glassy carbon disk working electrode (diameter 3 mm), Pt auxiliary electrode, and a SCE reference electrode. The electrolyte solutions were 0.1 M tetrabutylammonium hexafluorophosphate in DMF. Solutions were purged with $N_2$ or $CO_2$ over 30 min before measurements. All reported potentials in this work are versus SCE.

**Fluorescence quenching.** A solution of photosensitizer was degassed by $N_2$ for 15 min in a sealed quartz cuvette with a septum cap. Different concentrations of BIH or catalyst were added to the solution of photosensitizer under $N_2$. The steady-state fluorescence for solution samples was measured by Duetta fluorescence and absorbance spectrometer. The excited-state lifetime ($\tau_0$) of CuPP and PP was measured with an FLS 920 fluorescence spectrometer (Edinburgh instruments), in which a picosecond pulsed diode laser ($\lambda = 472$ nm) (Edinburgh instruments EPL-470) was used as the excitation source. The quenching rate constant ($k_q$) was calculated by Stern–Volmer equation:

$$I_0/I \text{ or } \tau_0/\tau = 1 + k_q \times \tau_0 \times [Q] \tag{1}$$

where $I_0$ and $I$ represent the fluorescence intensity of photosensitizer in the absence and presence of quencher, $k_q$ is the quenching rate constant, $\tau_0$ is the lifetime of the photosensitizer, and $[Q]$ is the concentration of quencher.

**Photocatalytic $CO_2$ reduction.** Photocatalytic experiments were conducted in a closed scintillation vial with rubber plug and magnetic stirring. The headspace of the vial was 53.4 mL. A reaction mixture (5.0 mL) was bubbled with $CO_2$ for 25 min and then irradiated with a white LED light setup ($\lambda > 400$ nm, PCX-50B/50 C, Beijing Perfectlight Technology Co., Ltd.). The gaseous products were analyzed by Shimadzu GC-2014 gas chromatography equipped with Shimadzu Molecular Sieve 13 × 80/100 3.2 × 2.1 mm × 3.0 m and Porapak N 3.2 × 2.1 mm × 2.0-m columns. A thermal conductivity detector (TCD) was used to detect $H_2$ and a flame ionization detector (FID) with a methanizer was used to detect CO and other hydrocarbons. Nitrogen was used as the carrier gas. The oven temperature was kept at 60 °C. The TCD detector and injection port were kept at 100 and 200 ºC, respectively. $^{13}C$ isotopic labeling experiments were carried out in a $^{13}CO_2$ atmosphere and gas products were detected by GC–MS (Agilent 7890A-5975C). HPLC (HP1100, Hewlett Packard) was used to analyze the photolysis products in the solution.

**Quantum yield measurement.** The difference between the power of light passing through the blank (containing FeTDHPP and BIH) and through the sample (containing CuPP, FeTDHPP, and BIH) was used to calculate the light absorbed by the CuPP. The average intensity of the irradiation $P$ (W/cm$^2$) was measured with a FZ-A Power meter (Beijing Normal University Optical Instrument company). The monochromic light of 450 nm obtained using blue LED light setup ($\lambda = 450$ nm, PCX-50B/50C, Beijing Perfectlight Technology Co., Ltd.). Generally, the quantum yield for $CO_2$ reduction to CO was calculated after irradiation for 1 h at 450 nm using the following equation:

$$\Phi_{CO} = \frac{2 \times \text{number of the CO molecules}}{\text{number of incident photons}} \times 100\% \tag{2}$$

that is,

$$\Phi_{CO} = \frac{2 \times n(CO)}{I} \times 100\% \tag{3}$$

where $n(CO)$ is the number of molecules of CO produced, $I$ is the number of incident photons; the calculation formula of the incident photon number $I$ is as follows:

$$I = PSt\frac{\lambda}{hc} \tag{4}$$

where $S$ is the incident irradiation area ($S = 0.785\ cm^2$), $t$ is the irradiation time, $\lambda$ is the incident wavelength, $h$ is the Plank constant ($6.626 \times 10^{-34}\ J \cdot s$), and $c$ is the speed of light ($3 \times 10^8\ m\ s^{-1}$)

$$\Phi = \frac{2 \times n \times N_A}{PSt \times \frac{\lambda}{hc}} \times 100\% \qquad (5)$$

where $N_A$ is the Avogadro constant ($6.02 \times 10^{23}\ mol^{-1}$).

## Data availability

The data that support the findings of this study are available from the corresponding author on reasonable request. The X-ray crystallographic coordinates for structures reported in this study have been deposited at the Cambridge Crystallographic Data Centre (CCDC), under deposition numbers 2017326. These data can be obtained free of charge from The Cambridge Crystallographic Data Centre via www.ccdc.cam.ac.uk/data_request/cif. Source data are provided with this paper.

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

## Acknowledgements

We are grateful for the financial support provided by Sun Yat-sen University. We thank Richard Eisenberg and Graham de Ruiter for helpful discussions.

## Author contributions

Z.H. supervised the project. H.Y. and Z.H. designed the experiments. H.Y. performed the synthesis of catalyst and photosensitizer and evaluated $CO_2$ reduction reactions. B.C. and J.L. performed the synthesis of BIH. L.J. contributed to the analysis of X-ray diffraction data. All authors analyzed the data. H.Y. and Z.H. prepared the paper.

## Competing interests

The authors declare no competing interests.
