## [Peer Review File · Nature Communications]

REVIEWER COMMENTS

Reviewer #1 (Remarks to the Author):

Review for Nature Commun. 273308_0

The full article of Prof. Han and co-worker is a beautiful report on photocatalytic CO₂ reduction with a system based on earth-abundant materials. In particular, a new photosensitizer was prepared from Cu(II) salt and purpurin ligand and its activity was evaluated in combination with an iron porphyrin. The results of the photocatalytic CO₂ reduction are astonishing, and they deserve publication by this journal. Nevertheless, major revisions are necessary prior publication, since some aspects are not well clarified, as well as a deeper discussion is missing. In particular, the authors should address the following points:

1) The new Cu(II)-purpurin (CuPP) has been characterized by X-ray, ESI-Mass and NMR spectroscopy, and elemental analysis. The authors report only the peaks of the ¹H NMR, while the integration of the signals should be reported as well (as they did for BIH and the porphyrin). Moreover, is there a reason why ¹³C NMR was not performed? Furthermore, in the Cif file only one tetrabutylammonium (TBA) counterion is shown. Why? Should not they be two TBA cations?

2) The photophysical characterization of (CuPP) is not complete. Since it is a new compound, its emission should be reported and compared also with the one of the free purpurin. What are the photoluminescence quantum yield and the radiative and nonradiative constant? Is the excitation spectrum comparable with the absorption spectrum? What is the nature of the excited state? Is it ligand centred? What is the effect of the Cu(II) ion? Comparison with other purpurin-chelated metal cores can be also done (for example see Photochem. Photobiol. Sci., 2011, 10, 1249–1254). This is of utmost importance especially if the authors target to “provide new perspectives for the rational design of efficient but low-cost photosensitizers for solar fuel production” (as the authors write at the end of the manuscript).

3) Regarding the lifetime experiment, in the experimental part should be reported which excitation source and excitation wavelength was used.

4) How do the authors explain the disappearance of some reduction processes (between -1.2 and -1.6V) when the CV under CO₂ atmosphere is compared with the one done under N₂? Why the first reduction (ca. -1 V) was not taken into consideration?

5) With a full photophysical and electrochemical characterization, the authors should report the redox potentials of the excited state. These values (E_{ox}* and E_{red}*) are used also to verify the free energy of the photoinduced electron transfer process (ΔG). In particular E_{red}* is important since the photosensitizer is expected to undergo reductive quenching by BIH. Therefore, the oxidation potential of BIH in the same solvent as the photocatalytic reactions (DMF) has to be reported as well.

6) In figure 4 the time-dependent evolution of CO with different combination of catalytic component is shown. In Fig.4a, the generation of CO by only CuPP is shown, and the authors write in the text that “the rate of CO evolution decreases dramatically after one hour. The disappearance of color of the reaction mixture after irradiation indicates the decomposition of CuPP”. If this is the case, that CuPP decomposes after one hour, then why this is not the case when the catalyst is present? What is the origin of such decomposition? Stability tests are shown in the supplementary figure (suppl.Fig. 10) where the catalytic activity of the system was tested for hours and only after ca. 20h addition of fresh CuPP was done to have a small increase of the total CO.

7) Later on in the manuscript, the authors write “absorption of the reaction mixture containing CuPP, FeTDHPP, and BIH in DMF was monitored by UV/vis spectroscopy during photolysis (Supplementary Fig. 11). The intensity of the absorption band at 400 nm, which corresponds to the reductions of PP ligands³⁴, decreases gradually on the course of irradiation. This confirms that CuPP decomposes during CO₂ reduction”. Then it is not clear what the species that act as photosensitizer is. In fact, if the absorption bands at 536 nm and 566 nm disappear already during the first 30 minutes and the band at 400 could be due to the reduced species of CuPP? In fact, if this experiment should prove the decomposition of the CuPP, then we do not know what is the species formed after its reductive quenching. Moreover, in the cited paper (reference 34: Dalton Trans. 2019, 48, 9596)the purpurin is used as photosensitizer: the absorption spectra of a similar experiment is shown to prove that the

reduced purpurin is shown, and this is comparable to the absorption spectra shown in Suppl. Fig. 11 of this manuscript. Notably, the spectrum of purpurin in DMF reported by the reference has also the bands at 530 nm and 560 nm. Can the authors comment on that?

8) The authors reveal the formation of some formic acid when N₂ atmosphere is used instead of CO₂. What process might generate formic acid? Does it come from the decomposition of some species in solution? Is there anything known in the literature?

9) The authors evaluate that the system is homogeneous. However, when they write "These control experiments that metal colloids are not responsible for CO₂ reduction in the CuPP/FeTDHPP/BIH system" is not clear if they see any metal colloids or macroscopic particles in solution. Which metal colloids?

10) The authors suggest a reaction mechanism in Fig. 7 and they discuss it in the text under the paragraph "mechanistic studies". The proposed mechanism shows that 3 electron reduction processes have to occur before the species "CuPP5-" absorbs light and its excited state "CuPP5-*" undergoes a reductive quenching by BIH. How these first 3 reduction processes occur? From which species CuPP is reduced? Further, why is the CuPP5- species that absorbs light and not the CuPP in ground state? In the text is written: "reduction of 1CuPP2-* by BIH is expected to be the first electron transfer step. Based on the electrochemical studies (Fig. 3a), CuPP undergoes up to four reductions to get to a CuPP6- species (Fig. 7). Indeed, the UV/vis spectrum of CuPP shows that a yellow species with absorption band at 400 nm is generated under visible light irradiation (Supplementary Fig. 11), which is consistent with double reductions of each PP ligands³⁴." This is not the same as the cycle reported in Fig. 7. The species that absorbs at 400nm is again reported to be the reduced species of purpurin, which is to be clarified as well (see my previous point 7).

11) Spectroelectrochemical experiments of the CuPP species might be helpful to investigate the correlation between the changes in the absorption spectra and the reduced species in solution.

12) The "Discussion" paragraph is more a "conclusion" one.

13) In the experimental part, the author write that CO was determined by Flame Ionization detector. However CO cannot be detectable by FID, unless a methanizer is used in line prior the detector. Further the carrier gas of the experiment has also to be reported.

14) In general, I find the absence of tables confusing. In fact, tables are very helpful for the readers to have a quick overview of the data. I suggest to add a table for the photophysical and electrochemical properties as well as one table with the results of the photocatalytic experiments. Control experiments in absence of BIH have to be done as well.

Reviewer #2 (Remarks to the Author):

This manuscript reports a highly active visible light-driven catalytic system for the reduction of CO₂ to CO. The catalytic system consists of a copper purpurin complex (CuPP) as photosensitizer, a chloroiron(III) porphyrin complex (FeTDHPP) as catalyst and BIH as sacrificial reductant. A TON of up to 16100 for CO production is achieved, with 95% selectivity, which apparently is the best result for photocatalytic CO₂ reduction with noble-metal-free catalytic systems. The use of a Cu purpurin complex as photosensitizer is a novel idea, the complex itself can also function as the catalyst, although the TON is low. This work may represent a significant advance in CO₂ reduction. However, there are certain issues which the authors should address:

1. Although a high turnover number (TON) of 16100 is achieved with this catalytic system, a very low Fe catalyst concentration of 0.2 micromolar is used. Such a low catalytic concentration is conceptually not very useful, since the total amount of product would be too small to be of any practical use even though the TON is high. The authors should try higher catalytic concentrations, probably up to 100 micromolar and report the TONs and product amounts as a function of catalyst concentration.
2. The authors reported that the performance of the CuPP complex is much better than PP using the Fe porphyrin as catalyst. Is this a general phenomenon? In supplementary table 2, four catalytic systems using purpurin are listed. It would be nice if the authors can use their CuPP complex as photosensitizer for at least one of these system to demonstrate the general superiority of CuPP over

PP.

3. Also in supplementary table 2, the catalyst concentrations of the various systems should be given in order to have a fair comparison of the TONs. Since the TON is defined by the amount of product divided by the amount of catalyst, a catalytic system with low catalyst concentration would tend to have a highly TON than one with high catalyst concentration. Hence, it is more fair to compare TONs using the same catalyst concentrations.

4. The CV of CuPP displays four irreversible reduction waves at -1.05, -1.50, and -1.69, -1.75 V vs SCE, suggesting that the reduced species of CuPP (CuPP3-, CuPP4-, CuPP5- and CuPP6-) may not be stable. Can the authors comment on this. Have the authors done repetitive scanning for the complex?

5. In fig 3b, the CV response of CuPP at -1.89 V vs. SCE under CO₂ atmosphere should not be simply attributed to the catalytic wave for CO₂ reduction. Reduction of protons or the Cu complex can also result in the formation of the wave. Electrolysis should be carried out to verify that the wave results from the reduction of CO₂.

6. In Fig. 7, CuPP(6-) is proposed as the active catalyst. Can the authors speculate on the nature of this species? What is the oxidation state of Cu? What is the nature of the PP ligand? Is it simply a radical anion, are the carbonyl groups still intact?

Reviewer #3 (Remarks to the Author):

This manuscript describes the photocatalytic conversion of CO₂ to CO in a multi component system containing a Cu-based photosensitizer and a Fe porphyrin catalyst in DMF. Thus, the authors present a system which is based only on earth-abundant elements, with very good efficiency and selectivity for photocatalytic reduction of CO₂ to CO. Even though the system performs in organic solvent (and not in water) and therefore the high selectivity over proton reduction is expected, the notable aspect of the current study is the very good efficiency of the system, which is among the best reported to date for such systems. The other notable aspect of the study is the Cu-based photosensitizer.

This is a very nice study that includes a substantial amount of data, including photochemical, and electrochemical analyses. The work appears to be well done. The paper is well written, the experiments have been well performed with a range of techniques being employed. The mercury poisoning experiments are very convincing, together with all the other control experiments and the isotopic labeling experiments. The conclusions are well supported by the data presented. The results of this work are of interest in the AP community and to the larger research community as they do open an effective approach for the rational design of efficient chromophores in energy conversion systems.

I recommend publication after the following points are addressed.

- Better supporting the assignment of the UV-vis spectrum features of the CuPP photosensitizer (PS) based on literature or TD-DFT calculations. Did the authors do or do they plan to do TD-DFT calculations for this compound?

- A more in-depth study of the photoluminescence properties of the CuPP would also be of interest. Comparisons of emission spectra of PP and CuPP obtained at different excitation wavelengths, in solution and in solid state, at room temperature and at low temperature would bring important information on the photophysical properties of the CuPP. I understand that it may be considered beyond the scope of this paper, but as a new PS is reported, information on its photoluminescence properties is of interest. In relation to this, what type of excited state do the authors propose for the CuPP? What are their thoughts about the geometry of the CuPP in the excited state?

- With respect to the electrochemical data, the reduction events in the CV of CuPP are difficult to

analyse. How did the authors establish how many electrons each reduction is? Were also square-wave voltammetry experiments done for CuPP and PP? In addition, as CuPP is a new compound, showing the oxidation part of the electrochemical analysis of the CuPP vs PP (at least in the Supplementary Information) would be of interest too.

- It will be interesting to see the profile of the TOF (at least in Supplementary Information) for the experiments shown in Fig. 4.
- For the experiment shown in Fig. 4b, there is an induction time of almost 1 h, which is attributed to the low concentration of BIH. Can the decomposition of the CuPP be ruled out during this long induction time?
- Did the authors perform a control experiment with using Cu salt + PP instead of CuPP?
- How was the concentration of CuPP chosen? Did the authors perform experiments at different CuPP concentrations?
- Did the authors perform any quantum yield measurements for this system?
- It is stated 'Cu-O distances, electronic charge balancing, and paramagnetic proton resonances in the ¹H NMR spectra (Supplementary Fig. 2) are all consistent with the assignment of a Cu(II) center.' However no values and no references are given for the Cu-O distances. The distances Cu-O in CuPP are reported only in SI. It would be of interest to be given in the manuscript as well. In addition, a CCDC search reveals that there are several reports of the solid state structures of Cu catecholates, 1,2-naphtoquinones (e.g., CEJBUP, BAKPUY). Comparison in term of bond lengths with these structures would be of interest as well.
- The authors state in the Conclusion: 'For example, the CuPP/FeTDHPP system achieves over 16100 turnovers of CO with a maximum TOF of 7650 h⁻¹, which is 2 orders of magnitude more than a reported Ir(ppy)₃/FeTDHPP system (TON = 140 in 55 h)³¹, and shows an almost 4-fold increase of a [Ru(bpy)₃]²⁺/CoTPPS system (TON = 4000, TOF_{max} = 2400 h⁻¹)⁵¹.' The comparison is very pertinent with respect to the Ir(ppy)₃/FeTDHPP system which is a system in organic solvent. However a distinction should be made with respect to the [Ru(bpy)₃]²⁺/CoTPPS system, which is a system performing in water.
- In the Supplementary table 2 and Supplementary table 3, it will be of interest to add the solvent in which the system performs.
- 'Supplementary Figure 1. ESI-MS spectra of Na₂Cu(PP)₂ in CH₃OH (negative ion mode). The most intense signals was at m/z ([M-Na⁺] -) = 594.22 (calcd: 593.96).' There is an important difference between the experimental and the calculated m/z. In addition, the comparison between the experimental isotopic pattern and the calculated isotopic pattern should also be presented. Did the authors also perform the MS for the Cu(PP)₂(TBA)₂?
- 'Supplementary Figure 11. UV-Vis absorption spectra of systems containing 5.0 μM CuPP, 0.2 μM FeTDHPP, and 10 mM BIH upon irradiation with white LED light.' To show a zoom on the region 450 – 650 nm would be of interest too.
- Have CH₄ also been quantified in this system?
- Could the CuPP and the catalyst also have favourable electrostatic interactions? Could this play a role in the enhanced performance?

Minor points:

- As a suggestion, the paper would benefit by moving the Fig. 5 (Illustrative scheme of the photocatalytic CO₂ reduction system investigated in this study) at the beginning of the manuscript before the Results, where the authors present the system under discussion in this work.
- It is written 'The Yield 21.8%'. Reporting the overall yield of this type of synthesis with such precision is difficult. The value should be 22%.
- In Supplementary table 1, in the footnote caption, the description of entries 6 and 7 are missing.
- The excitation wavelength should be mentioned in the Supplementary Figure 13 and the Supplementary Figure 14.
- In the manuscript as section title, it is written Discussion instead of Conclusion.

Reviewer #1

We would like to thank Reviewer 1 for his/her comments and suggestions. Each point is addressed below.

“The full article of Prof. Han and co-worker is a beautiful report on photocatalytic CO₂ reduction with a system based on earth-abundant materials. In particular, a new photosensitizer was prepared from Cu(II) salt and purpurin ligand and its activity was evaluated in combination with an iron porphyrin. The results of the photocatalytic CO₂ reduction are astonishing, and they deserve publication by this journal. Nevertheless, major revisions are necessary prior publication, since some aspects are not well clarified, as well as a deeper discussion is missing. In particular, the authors should address the following points”

Thank you very much for your insightful reading of our manuscript and for your helpful comments. We have taken the referees' advice into account and performed a significant amount of further experiments. We have rewritten the manuscript to have a deeper discussion of mechanistic study of the photocatalytic systems.

(1) “The new Cu(II)-purpurin (CuPP) has been characterized by X-ray, MS and NMR spectroscopy, and elemental analysis. The authors report only the peaks of the ¹H NMR, while the integration of the signals should be reported as well (as they did for BIH and the porphyrin). Moreover, is there a reason why ¹³C NMR was not performed? Furthermore, in the Cif file only one tetrabutylammonium (TBA) counterion is shown. Why? Should not they be two TBA cations?”

We thank the referee for pointing this out. We have taken the ¹H NMR spectra of CuPP in both *d*₆-DMSO and CD₃CN and included the integrations for the spectra (see following figures). From the integrations, there are total five protons at the aromatic region. However, due to the paramagnetic nature of the Cu(II) compound, we could not assign these peaks, which is a common problem observed for paramagnetic metal complexes.

Supplementary Figure 3. ¹H NMR spectrum of CuPP in *d*₆-DMSO.

Supplementary Figure 4. ^1H NMR spectrum of CuPP in CD_3CN .

We have previously tried taking the ^{13}C NMR spectrum of CuPP, however, we could not detect any meaningful signal associated with the complex, presumably due to its paramagnetic nature. Even though with 3000 scans, we could only observe 4 peaks which are from the TBA cations:

Supplementary Figure 5. ^{13}C NMR spectrum of CuPP in d_6 -DMSO.

Regarding to the cif. file, if using Mercury software to open, we also experience a problem displaying the whole molecule using the default setting. By checking the “Asymmetric Unit” tick box, it displays half of the molecule correctly. Or if using the “Packing” function, it clearly shows there are two molecules (four TBA cations) in each unit cell. We have included a check cif file along with our submission, and the “Moiety formula” shows there are two TBA in the molecule as well.

(2) “*The photophysical characterization of (CuPP) is not complete. Since it is a new compound, its emission should be reported and compared also with the one of the free purpurin. What are the photoluminescence quantum yield and the radiative and nonradiative constant? Is the excitation spectrum comparable with the absorption spectrum? What is the nature of the excited state? Is it ligand centred? What is the effect of the Cu(II) ion? Comparison with other purpurin-chelated metal cores can be also done (for example see Photochem. Photobiol. Sci., 2011, 10, 1249–1254). This is of utmost importance especially if the authors target to “provide new perspectives for the rational*

design of efficient but low-cost photosensitizers for solar fuel production” (as the authors write at the end of the manuscript.”

We thank the referee for the suggestion. We have included additional data to characterize the photophysical properties of CuPP. We have added the following figures and table to the MS and SI (Fig. 3b-c, Supplementary Fig. 11-14 and 35-36, and Supplementary Table 2). The photoluminescence quantum yields for PP and CuPP are 2.7×10^{-2} and 8.2×10^{-3} respectively. This is different from the observation reported in the literature (*Photochem. Photobiol. Sci.*, **2011**, *10*, 1249-1254) in which adding Al(III) to PP results in a significant increase of the fluorescence intensity. This is probably due to the nature of the emissions are different for the CuPP and Al-PP compounds. The radiative and nonradiative constants for CuPP and PP have also been included in Supplementary Table 2.

The excitation spectrum of PP is comparable with the absorption spectrum, which is in agreement with the literature. However, the absorption and excitation spectra for CuPP are very different (see following figures), suggesting a structural change in its excited state. Because the absorption spectrum of CuPP has very similar feature compared to the di-anion form of PP (*J. Phys. Org. Chem.* **2000**, *13* (3), 141-150), the absorption in the visible region for CuPP is assigned to the charge transfer bands on the PP ligands. The incorporation of Cu(II) enhances this charge transfer process by showing a much higher molar extinction coefficient. The emission of CuPP is most intense at 375 nm excitation, which is similar to the MLCT and LMCT bands reported for Cu complexes (*Angew. Chem.* **2013**, *125*, 437-441; *J. Am. Chem. Soc.* **1974**, *96*, 6868-6873), suggesting this emission is probably related to a LMCT band at around 375 nm (overlapped with the transitions on the PP ligands). Thus, the emission of CuPP is from the relaxation of a Cu(I) species generated from LMCT.

We have added the following paragraph to the “Preparation and characterization of CuPP” section to describe the photophysical properties of CuPP:

“Because the UV-vis spectrum of CuPP exhibits identical feature as compared to the di-anion form of PP in the visible region⁵⁰, the absorption band at 566 nm is assigned to the charge transfer band on the PP ligands. The incorporation of Cu(II) enhances this charge transfer process by showing a much higher molar extinction coefficient (Supplementary Fig. 10). The photoluminescence quantum yield for CuPP (8.2×10^{-3}) is lower than that of PP (2.7×10^{-2}) (Supplementary Table 2). Instead, Romani and co-workers observed an increase of the fluorescence intensity when adding Al(III) to PP⁵¹. The distinct luminescent behavior of CuPP and Al-PP is probably due to the reason that the nature of emissions are based on different excited states. Excitation-emission spectra (Fig. 3b-c) show that the emission of CuPP is most intense when excited at 375 nm, which is similar to the charge transfer regions reported for Cu complexes^{52,53}. These data suggest that the emission of CuPP comes from a Cu(I) excited state generated from a LMCT process. The excitation spectrum of PP is comparable with the absorption spectrum (Supplementary Fig. 11a), which is in agreement with the literature⁵⁴. In contrast, the distinct absorption and excitation spectra of CuPP (Supplementary Fig. 11b) suggest a significantly structural change in its excited state, which is consistent with the observation that Cu(I) tends to adopt a tetrahedral geometry⁵⁵.”

Supplementary Table 2 Photophysical parameters of PP and CuPP at 298 K in DMF solution

Sample	Medium	Absorbance		Fluorescence				
		λ_{\max} (nm)	$\epsilon_{\lambda,\max}$ ($M^{-1} \text{ cm}^{-1}$)	λ_{\max} (nm)	Φ_F^a	τ (ns)	$\kappa_F/10^7$ (s^{-1})	$\kappa_{NF}/10^9$ (s^{-1})
PP	DMF	478	9940	582 ^b	2.7×10^{-2}	1.1	2.5	0.88
CuPP	DMF	566	38530	693 ^c	8.2×10^{-3}	0.98	0.84	1.01

^a Measured under N_2 , by absolute method using an integrating sphere, error 1-20 %. ^b $\lambda_{\text{exc}} = 484$ nm; ^c $\lambda_{\text{exc}} = 540$ nm.

**Fig. 3.** Excitation and emission spectra of PP (b) and CuPP (c) at different wavelengths at 298 K in DMF.**Supplementary Figure 11.** Normalized excitation and absorbance spectra of PP (a) and CuPP (b) at 298 K in DMF.**Supplementary Figure 12.** Normalized absorbance and emission spectra of PP (a, λ_{exc} : 484 nm) and CuPP (b, λ_{exc} : 540 nm) at 298 K in DMF.

(3) “Regarding the lifetime experiment, in the experimental part should be reported which excitation source and excitation wavelength was used.”

We regret the mistake and have added the excitation source and excitation wavelength to the experimental section as follows: “A picosecond pulsed diode laser ($\lambda = 472$ nm) (Edinburgh instruments EPL-470) was used as the excitation source.”

(4) “How do the authors explain the disappearance of some reduction processes (between -1.2 and -1.6V) when the CV under CO₂ atmosphere is compared with the one done under N₂? Why the first reduction (ca. -1 V) was not taken into consideration?”

The referee brings up a very good point in this comment. According to the literature (J. *Electroanal. Chem.* **1992**, 322, 383-389), the CV of 9,10-anthracenediones changes significantly under CO₂, which is due to the complexation of CO₂ with the two quinone moieties to form carbonates at nearly the same potential. In Figure 4c, we observed that both PP and CuPP exhibit quasi-reversible reduction waves at similar potentials, suggesting that they both undergo complexation of CO₂ to generate carbonates.

We have revised the “Electrochemical analysis of CuPP” section to include description of this observation as follows:

“Figure 4 shows the cyclic voltammograms (CVs) of CuPP and PP in DMF solution containing 0.1 M tetrabutylammonium hexafluorophosphate (TBAPF₆). Indeed, the incorporation of a Cu(II) center into PP creates additional reduction events. CuPP displays three irreversible reduction potentials at -1.05, -1.50, and -1.69 V vs SCE. The most negative redox couple of CuPP at -1.75 V is 540 mV more negative than the PP²⁻/PP⁻ couple (Fig. 4a)³². Square wave voltammetry (SWV) reveals that there are four successive reductions, which correspond to 2, 1, 1, and 2 electron processes (Supplementary Fig. 17).

To explore the electron requirements for photocatalytic systems, further CV experiments were performed to study the electrocatalytic activity for CO₂ reduction. Under an atmosphere of CO₂, CuPP shows a quasi-reversible reduction wave at -0.97 V and a catalytic wave which starts to appear below -1.89 V as compared to no observation of current enhancement in N₂ (Fig. 4b), suggesting that CuPP could act as a CO₂ reduction catalyst. CV control experiments for PP and electrolyte solution show little current enhancement at the same potentials in the presence of CO₂ (Fig. 4c). The integrals of SWV show a total of four electron reduction at -1.13V prior to the catalytic current (Supplementary Fig. 17). The different electrochemical profiles observed for CuPP in the presence of CO₂ and N₂ indicate that the four quinone moieties on the PP ligands undergo four electron reductions and four complexations of CO₂ to generate carbonates. In fact, a similar observation for 9,10-anthracenediones has been reported in the literature⁵⁶. Controlled potential coulometry experiments, performed at -1.89 V (vs SCE) in CO₂-saturated DMF, show that CO is the main gas product with a faradaic yield of 17% and only negligible amount of H₂ generated (Supplementary Fig. 20). In a control experiment without CuPP, there is little CO or H₂ produced. These data confirm that the observed current enhancement of CuPP corresponds to CO₂ reduction not proton reduction.”

(5) “With a full photophysical and electrochemical characterization, the authors should report the redox potentials of the excited state. These values (E_{ox}^* and E_{red}^*) are used also to verify the free energy of the photoinduced electron transfer process (ΔG). In particular E_{red}^* is important since the photosensitizer is expected to undergo reductive quenching by BIH. Therefore, the oxidation potential of BIH in the same solvent as the photocatalytic reactions (DMF) has to be reported as well.”

We thank the referee for the suggestion. We have calculated the redox potentials of the excited states for CuPP and PP based on photophysical and electrochemical measurements. We have added the following CV data for BIH and added Table 3 to the SI to include the E_{ox}^* , E_{red}^* , and ΔG values. We have also added discussion of the data in the “Mechanistic studies” section (please see our reply to question 10).

Supplementary Table 3 Thermodynamic driving force for electron transfer of photocatalytic systems

Photosensitizers	E_{red} / V	$E_{0,0} / eV$	E_{red}^* / V	E_{ox}^* / V	$\Delta G / eV$
PP	-1.21	2.34	1.13	-1.47	-0.8
CuPP	-1.75	2.12	0.37	-1.61	-0.04

$E_{0,0}$ values were determined from the intersection of the normalized absorption and emission spectra of the CuPP, in CO_2 -saturated DMF solution, and converted to eV (Reference: *Chem. Eur. J.* **2013**, *19*, 15972-15978). The ground state redox potentials (E_{ox} and E_{red}) were measured by electrochemical methods (CVs). The excited state redox potentials were obtained as follows: ESOP (Excited State Oxidation Potential) = $E_{ox}(CuPP^*) = E_{ox} - E_{0,0}$; ESRP (Excited State Reduction Potential) = $E_{red}(CuPP^*) = E_{red} + E_{0,0}$. The thermodynamic driving force for electron transfer were calculated from Rehm-Weller equation: the difference between reduction potential of excited state of photosensitizer and oxidation potential of BIH as sacrificial reagent. ($\Delta G = E_{(D+D)}^0 - E_{(A/A-)}^0 - E_{0,0} - e^2/\epsilon d$). The last term which represents the coulombic attraction energy was neglected because of small contribution to the overall energy. Therefore, the equation was simplified to $\Delta G = E_{ox}(BIH) - E_{red}^*(CuPP)$ where $E_{ox}(BIH)$ was +0.33 V (vs SCE). Potentials are given versus SCE.

Supplementary Figure 15. CV of BIH (5 mM) in DMF containing 0.1M TBAPF₆ at 0.1 V/s scan rate.

(6) “In figure 4 the time-dependent evolution of CO with different combination of catalytic component is shown. In Fig. 4a, the generation of CO by only CuPP is shown, and the authors write in the text that “the rate of CO evolution decreases dramatically after one hour. The disappearance of color of the reaction mixture after irradiation indicates the decomposition of CuPP”. If this is the case, that CuPP decomposes after one hour, then why this is not the case when the catalyst is present? What is the origin of such decomposition? Stability tests are shown in the supplementary figure (suppl. Fig. 10) where the catalytic activity of the system was tested for hours and only after ca. 20h addition of fresh CuPP was done to have a small increase of the total CO.”

We thank the referee for the questions. We have performed experiments monitoring the reactions in real-time using UV-vis spectroscopy. These spectra were taken from the exact photolysis conditions as Fig. 5a and 5c (previous Fig. 4a and 4c) to reflect the actual spectral changes on the course of CO production (see following figures). We have revised our manuscript to show that at high [BIH], the reduced CuPP specie (at 400 nm) is the active photosensitizer for CO production (please also see our reply to question 10). To confirm that this species is indeed a reduced CuPP and is not a decomposition product, we could recover most of the CuPP during photolysis by opening the reaction vials to the air (please see our reply to question 7). Furthermore, we observe that this reduced species lasts much longer in the presence of catalyst from the UV-vis experiments. Without a catalyst, most of the reduced species goes away at ~1h (Supplementary Figure 29). In the presence of catalyst, this species only starts to decrease at ~2 h and almost completely disappears at ~9 h (Fig. 8b). This observation is consistent with some previous reports (*J. Am. Chem. Soc.* **2013**, *135*, 14659-14669; *J. Am. Chem. Soc.* **2010**, *132*, 15480-15483) which have showed that the presence of catalyst could stabilize the system by reducing the lifetime of unstable organic PS⁻ in solution.

Supplementary Figure 29. UV-Vis absorption spectra of systems containing: (a) 0.1 mM CuPP, and 100 mM BIH upon irradiation with white LED light in a 2 mm path length of quartz cuvette (dilution factor of 5).

Fig. 8 UV-vis absorption spectra of systems containing (b) 0.1 mM CuPP, 0.2 μM FeTDHPP and 100 mM BIH upon irradiation with white LED light. Solutions for (b) at different times were transferred and diluted 5 times with DMF to a quartz cuvette (2 mm path length) under N₂.

(7) “Later on in the manuscript, the authors write “absorption of the reaction mixture containing CuPP, FeTDHPP, and BIH in DMF was monitored by UV/vis spectroscopy during photolysis (Supplementary Fig. 11). The intensity of the absorption band at 400 nm, which corresponds to the reductions of PP ligands³⁴, decreases gradually on the course of irradiation. This confirms that CuPP decomposes during CO₂ reduction”. Then it is not clear what the species that act as photosensitizer is. In fact, if the absorption bands at 536 nm and 566 nm disappear already during the first 30 minutes and the band at 400 could be due to the reduced species of CuPP? In fact, if this experiment should prove the decomposition of the CuPP, then we do not know what is the species formed after its reductive quenching. Moreover, in the cited paper (reference 34: Dalton Trans. 2019, 48, 9596) the purpurin is used as photosensitizer: the absorption spectra of a similar experiment is shown to prove that the reduced purpurin is shown, and this is comparable to the absorption spectra shown in Suppl. Fig. 11 of this manuscript. Notably, the spectrum of purpurin in DMF reported by the reference has also the bands at 530 nm and 560 nm. Can the authors comment on that?”

We thank the referee for the questions. We have performed UV-vis experiments to confirm that the 400 nm species is indeed a reduced species of CuPP and is not a decomposition product generated during photolysis. PP shows a different absorption spectrum compared to CuPP in BIH (see following left figure). This is consistent with a di-anion form of PP in basic solution reported in the literature (*J. Phys. Org. Chem.* **2000**, 13 (3), 141-150). We could recover most of the CuPP from the species at 400 nm by opening the reaction vials to the air (following right figure). Thus, CuPP is an active photosensitizer for CO production in our study. We have added the following sentences to the “Mechanistic studies” section to describe this observation:

“At high [BIH] (100 mM), a yellow species with absorption band at 400 nm is generated under visible light irradiation (Fig. 8b). When the reaction mixture is exposed to the air, the solution returns to purple color and over 80% of the original CuPP is recovered based on UV-vis spectra (Supplementary Fig. 34).”

Supplementary Figure 34. UV-vis absorption spectra of systems containing: (a) 0.1 mM CuPP (red) or 0.2 mM PP (black), 0.04 μ M FeTDHPP and 20 mM BIH (spectra taken with dilution factor of 5); (b) 0.1 mM CuPP, 0.2 μ M FeTDHPP and 100 mM BIH before irradiation (black), irradiation for 10 minutes (red) then bubbled with air (blue) (spectra taken with dilution factor of 50). Condition: (a) 2 mm path length quartz cuvette; (b) 1 cm path length quartz cuvette.

(8) *“The authors reveal the formation of some formic acid when N₂ atmosphere is used instead of CO₂. What process might generate formic acid? Does it come from the decomposition of some species in solution? Is there anything known in the literature?”*

We thank the referee for the questions. Formic acid could come from hydrolysis of DMF. This phenomenon has been reported in the literature (*Inorg. Chem.* **2014**, 53, 3326). We have added this reference to the manuscript and revised the text as follows:

“In the solution, 1.4 μmol formic acid was quantified by high performance liquid chromatography (HPLC). However, a similar amount (1.5 μmol) of formic acid was detected from the experiment performed under N₂, suggesting that formic acid is not generated from CO₂ reduction. In fact, formic acid generated from hydrolysis of DMF has been previously reported⁵⁸.”

(9) *“The authors evaluate that the system is homogeneous. However, when they write “These control experiments suggest that metal colloids are not responsible for CO₂ reduction in the CuPP/FeTDHPP/BIH system” is not clear if they see any metal colloids or macroscopic particles in solution. Which metal colloids?”*

We regret the confusion and have revised the appropriate sentence to now read: *“These control experiments suggest that there are no Cu colloids generated and the CuPP/FeTDHPP/BIH system stays homogeneous during photocatalytic CO₂ reduction.”*

(10) *“The authors suggest a reaction mechanism in Fig. 7 and they discuss it in the text under the paragraph “mechanistic studies”. The proposed mechanism shows that 3 electron reduction processes have to occur before the species “CuPP5 - “absorbs light and its excited state “CuPP5-*” undergoes a reductive quenching by BIH. How these first 3 reduction processes occur? From which species CuPP is reduced? Further, why is the CuPP5- species that absorbs light and not the CuPP in ground state? In the text is written: “reduction of 1CuPP2-* by BIH is expected to be the first electron transfer step. Based on the electrochemical studies (Fig. 3a), CuPP undergoes up to four reductions to get to a CuPP6- species (Fig. 7). Indeed, the UV/vis spectrum of CuPP shows that a yellow species with absorption band at 400 nm is generated under visible light irradiation (Supplementary Fig. 11), which is consistent with double reductions of each PP ligands³⁴.” This is not the same as the cycle reported in Fig. 7. The species that absorbs at 400nm is again reported to be the reduced species of purpurin, which is to be clarified as well (see my previous point 7).”*

We thank the referee for the questions. We agree with the referee that our originally proposed mechanism was confusing. We have performed additional experiments to come up with a more complete picture of mechanism for CO₂ reduction. We have proposed a new mechanistic scheme, and revised the “Mechanistic studies” section, and added the following paragraph:

“CO generation was observed even at [BIH] as low as 0.1 mM. UV-vis spectra show no generation of additional absorption band in the visible region during CO₂ reduction (Fig. 8a), indicating CuPP is the active PS at this condition. Since [L₂Cu^{II}]⁻ (L = PP²⁻) has a much higher reduction potential (-1.05 V) than that of Fe(I)TDHPP (E_{1/2} = -1.55 V), CO₂ reduction has to proceed through oxidative quenching of L₂Cu^{II} (-1.61 V) (Supplementary Table 4) in the photochemical step (Fig. 7). At high*

[BIH] (100 mM), a yellow species with absorption band at 400 nm is generated under visible light irradiation (Fig. 8b). When the reaction mixture is exposed to the air, the solution returns to purple color and over 80% of the original CuPP is recovered based on UV-vis spectra (Supplementary Fig. 34). Electrochemical studies in the presence of CO₂ (Fig. 5b) suggest that CuPP could undergo four steps of reduction and complexation of CO₂ to generate a $[(\text{CO}_2)_2\text{L}_2\text{Cu}^{\text{II}}]^{4-}$ species (Fig. 7). In fact, PP has been shown to proceed a 2-electron reduction to generate a species at 398 nm during photocatalytic CO₂ reduction³⁴. Thus, we conclude that the intermediate with maximum absorption at 400 nm is a $[(\text{CO}_2)_2\text{L}_2\text{Cu}^{\text{II}}]^{4-}$ species. $[(\text{CO}_2)_2\text{L}_2\text{Cu}^{\text{I}}]^{5-}$ (-1.75V), generated by further reduction of $[(\text{CO}_2)_2\text{L}_2\text{Cu}^{\text{II}}]^{4-}$ through reductive quenching, is corresponding to electron transfer to the Fe catalyst (Fig. 7).”

Fig. 7 Proposed reaction scheme for photocatalytic CO₂ reduction

(11) “Spectroelectrochemical experiments of the CuPP species might be helpful to investigate the correlation between the changes in the absorption spectra and the reduced species in solution.”

We thank the referee for the suggestion. We have tried to perform spectroelectrochemical experiments using a three-electrode setup. We did observe absorption bands at the 400–450 nm region upon applying a negative potential. However, the reduced species generated decomposed rapidly on the Pt-mesh working electrode (see following figure) and we could not re-oxidize it back to CuPP. Thus, we do not yet have convincing evidence to identify the species generated from the spectroelectrochemical experiments. We are currently building a more sophisticated cell so that we could use other working electrodes for the study. Spectroelectrochemical study are certainly part of our goals in future studies. However, from the UV-vis experiments, we are convinced that the 400 nm absorption band generated during CO production is the reduced CuPP (please see our reply to question 7).

Fig. Spectroelectrochemical experiments of CuPP (20 μM) with controlled potential at -2.1 V (vs SCE) in DMF containing 100 mM TBAPF₆.

(12) *“The “Discussion” paragraph is more a “conclusion” one.”*

Thank you for pointing this imprecision out. Because Nature Communications does not have a separated section for “Conclusion”, we have changed the section title “Result” into “Result and Discussion” and deleted the section title “Discussion” for the last paragraph.

(13) *“In the experimental part, the author write that CO was determined by Flame Ionization detector. However CO cannot be detectable by FID, unless a methanizer is used in line prior the detector. Further the carrier gas of the experiment has also to be reported.”*

We thank the referee for pointing this out. Information about the detectors and carrier gas have been added to the experimental section. We have revised the appropriate sentences to now read, “A thermal conductivity detector (TCD) was used to detect H₂ and a flame ionization detector (FID) with a methanizer was used to detect CO and other hydrocarbons. Nitrogen was used as the carrier gas.”

(14) *“In general, I find the absence of tables confusing. In fact, tables are very helpful for the readers to have a quick overview of the data. I suggest to add a table for the photophysical and electrochemical properties as well as one table with the results of the photocatalytic experiments. Control experiments in absence of BIH have to be done as well.”*

We thank the referee for the suggestion. We have organized the photophysical, electrochemical, and photocatalytic data including the control experiments in the absence of BIH into the following tables:

Supplementary Table 2 Photophysical parameters of PP and CuPP at 298 K in DMF solution

Sample	Medium	Absorbance		Fluorescence				
		$\lambda_{\text{max}}(\text{nm})$	$\epsilon_{\lambda_{\text{max}}}(\text{M}^{-1} \text{cm}^{-1})$	$\lambda_{\text{max}}(\text{nm})$	$\Phi_{\text{F}}^{\text{a}}$	τ (ns)	$\kappa_{\text{F}}/10^7 (\text{s}^{-1})$	$\kappa_{\text{NF}}/10^9 (\text{s}^{-1})$
PP	DMF	478	9940	582 ^b	2.7×10^{-2}	1.1	2.5	0.88
CuPP	DMF	566	38530	693 ^c	8.2×10^{-3}	0.98	0.84	1.01

^aMeasured under N₂, by absolute method using an integrating sphere, error 1-20 %. ^b $\lambda_{exc} = 484$ nm; ^c $\lambda_{exc} = 540$ nm.

Supplementary Table 3 Thermodynamic driving force for electron transfer of photocatalytic systems

Photosensitizers	E_{red} / V	$E_{0,0} / eV$	E_{red}^* / V	E_{ox}^* / V	$\Delta G / eV$
PP	-1.21	2.34	1.13	-1.47	-0.8
CuPP	-1.75	2.12	0.37	-1.61	-0.04

E_{0-0} values were determined from the intersection of the normalized absorption and emission spectra of the CuPP, in CO₂-saturated DMF solution, and converted to eV (Reference: Chem. Eur. J. 2013, 19, 15972-15978). The ground state redox potentials (E_{ox} and E_{red}) were measured by electrochemical methods (CVs). The excited state redox potentials were obtained as follows: ESOP (Excited State Oxidation Potential) = $E_{ox}(CuPP^*) = E_{ox} - E_{0-0}$; ESRP (Excited State Reduction Potential) = $E_{red}(CuPP^*) = E_{red} + E_{0-0}$. The thermodynamic driving force for electron transfer were calculated from Rehm-Weller equation: the difference between reduction potential of excited state of photosensitizer and oxidation potential of BIH as sacrificial reagent. ($\Delta G = E^0_{(D+/D)} - E^0_{(A/A-)} - E_{0,0} - e^2/\epsilon d$). The last term which represents the columbic attraction energy was neglected because of small contribution to the overall energy. Therefore, the equation was simplified to $\Delta G = E_{ox}(BIH) - E_{red}^*(CuPP)$ where $E_{ox}(BIH)$ was +0.33 V (vs SCE). Potentials are given versus SCE.

Table 1 Control and other photocatalytic CO₂ reduction experiments to characterize the performance of the BIH/CuPP/FeTDHPP system in CO₂-saturated DMF solution

Entry	PS	[FeTDHPP] (μM)	[BIH] (mM)	Irradiation time (h)	CO (μmol)	H ₂ (μmol)	TON _{CO}	TOF _{CO} ^{max} (h ⁻¹)	TON _{H2}	SEL _{CO} (%)
1	PP	0.2	10	7	0.088	0	88	45	0	100
2	CuPP	0.2	10	7	4.779	0.27	4779	1356	270	95
3	CuPP	0.2	100	23	16.109	0.843	16109	7650	843	95
4	CuPP	0	100	1	2.2	0	4.4 ^a	4.3	0	100

^a Calculated by the equation $TON_{CO} = n(CO)/n(CuPP)$

Reviewer #2

We would like to thank Reviewer 2 for his/her comments and suggestions. Each point is addressed below.

“This manuscript reports a highly active visible light-driven catalytic system for the reduction of CO₂ to CO. The catalytic system consists of a copper purpurin complex (CuPP) as photosensitizer, a chloroiron(III) porphyrin complex (FeTDHPP) as catalyst and BIH as sacrificial reductant. A TON of up to 16100 for CO production is achieved, with 95% selectivity, which apparently is the best result for photocatalytic CO₂ reduction with noble-metal-free catalytic systems. The use of a Cu purpurin complex as photosensitizer is a novel idea, the complex itself can also function as the catalyst, although the TON is low. This work may represent a significant advance in CO₂ reduction. However, there are certain issues which the authors should address”

Thank you very much for your insightful reading of our manuscript and for your helpful comments. We have taken the referees' advice into account and performed a significant amount of further experiments. We have rewritten the manuscript to have a deeper discussion of the photocatalytic systems.

(1) “Although a high turnover number (TON) of 16100 is achieved with this catalytic system, a very low Fe catalyst concentration of 0.2 micromolar is used. Such a low catalytic concentration is conceptually not very useful, since the total amount of product would be too small to be of any practical use even though the TON is high. The authors should try higher catalytic concentrations, probably up to 100 micromolar and report the TONs and product amounts as a function of catalyst concentration.”

We thank the referee for the suggestion. We have carried out experiments to investigate the effect of catalyst concentration (up to 100 μ M) on catalysis. We have added the following paragraph to the “Photocatalytic CO₂ reduction” section to describe the effects of concentration of photosensitizer and catalyst:

“The activity of the photochemical systems were optimized further by varying the [CuPP] and [FeTDHPP]. At fixed concentrations of BIH (100 mM) and FeTDHPP (2 μ M), increasing the CuPP concentration increases the overall rate of CO production (Supplementary Fig. 23). However, the rate of CO generation does not increase above 0.1 mM concentration of CuPP. When the CuPP concentration is fixed at 0.1 mM and [FeTDHPP] is varied (Supplementary Fig. 24), the catalyst is most active (on a TON basis) at low concentrations. The total amount of CO evolved increases up to 40 μ M FeTDHPP, suggesting that the activity of the system becomes limited by electron transfer to the catalyst.”

We have added the following graph to the SI:

Supplementary Figure 24. TON (dot) and amounts (triangle) of CO after irradiation 23h of the photocatalytic CO₂ reduction experiments in CO₂-saturated DMF solution containing 0.1 mM CuPP and 100 mM BIH with varying amounts of FeTDHPP.

(2) “The authors reported that the performance of the CuPP complex is much better than PP using the Fe porphyrin as catalyst. Is this a general phenomenon? In supplementary table 2, four catalytic systems using purpurin are listed. It would be nice if the authors can use their CuPP complex as photosensitizer for at least one of these system to demonstrate the general superiority of CuPP over PP.”

Thank you for the suggestion. We have performed photocatalytic experiments using Co(qpy)Cl₂ as the catalyst for CO₂ reduction. We observed an almost 2-fold increase of activity when using CuPP as the PS (see following graph). The less pronounced increase of activity in this system compared with the BIH/CuPP/FeTDHPP system is probably due to Co(qpy)Cl₂ is less active than FeTDHPP. In this case, even though CuPP could promote electron transfer to the catalyst, the overall catalytic rate becomes limited by the activity of catalyst. We have added these results to the SI (Supplementary Figure 27).

Supplementary Figure 27. Photocatalytic CO₂ reduction in CO₂-saturated DMF solutions containing: 0.2 mM PP (black) or 0.1 mM CuPP (red), 2 μM Co(qpy)Cl₂ (qpy = 2,2':6',2":6",2'''-quaterpyridine) ((Reference: *J. Am. Chem. Soc.* **2016**, *138*, 9413-9416) and 30 mM BIH.

(3) *“Also in supplementary table 2, the catalyst concentrations of the various systems should be given in order to have a fair comparison of the TONs. Since the TON is defined by the amount of product divided by the amount of catalyst, a catalytic system with low catalyst concentration would tend to have a highly TON than one with high catalyst concentration. Hence, it is more fair to compare TONs using the same catalyst concentrations.”*

This is a good suggestion. We have added catalyst concentrations to the supplementary table 5-8 for comparison.

(4) *“The CV of CuPP displays four irreversible reduction waves at -1.05, -1.50, and -1.69, -1.75 V vs SCE, suggesting that the reduced species of CuPP (CuPP3-, CuPP4-, CuPP5-and CuPP6-) may not be stable. Can the authors comment on this. Have the authors done repetitive scanning for the complex?”*

We thank the referee for the question. We have performed CV experiments with multiple scans for CuPP (results included in Supplementary Fig. 16). The reduction waves are reproducible during these scans. The irreversible waves of CuPP are probably due to structural changes upon reductions. Related to the question, we observed that the absorption and excitation spectra of CuPP are very different (Supplementary Fig. 11), which again suggests a structural change in the Cu(I) excited state. We have added the following paragraph to the “Preparation and characterization of CuPP” section to discuss this point:

“Because the UV-vis spectrum of CuPP exhibits identical feature as compared to the di-anion form of PP in the visible region⁵⁰, the absorption band at 566 nm is assigned to the charge transfer band on the PP ligands. The incorporation of Cu(II) enhances this charge transfer process by showing a much higher molar extinction coefficient (Supplementary Fig. 10). The photoluminescence quantum yield for CuPP (8.2×10^{-3}) is lower than that of PP (2.7×10^{-2}) (Supplementary Table 2). Instead, Romani and co-workers observed an increase of the fluorescence intensity when adding Al(III) to PP⁵¹. The distinct luminescent behavior of CuPP and Al-PP is probably due to the reason that the nature of emissions are based on different excited states. Excitation-emission spectra (Fig. 3b-c) show that the emission of CuPP is most intense when excited at 375 nm, which is similar to the charge transfer regions reported for Cu complexes^{52,53}. These data suggest that the emission of CuPP comes from a Cu(I) excited state generated from a LMCT process. The excitation spectrum of PP is comparable with the absorption spectrum (Supplementary Fig. 11a), which is in agreement with the literature⁵⁴. In contrast, the distinct absorption and excitation spectra of CuPP (Supplementary Fig. 11b) suggest a significantly structural change in its excited state, which is consistent with the observation that Cu(I) tends to adopt a tetrahedral geometry⁵⁵.”

Supplementary Figure 16. CV of CuPP (1 mM) with multiple scans in DMF containing 0.1M TBAPF₆ at 0.1 V/s scan rate.

(5) “In fig 3b, the CV response of CuPP at -1.89 V vs. SCE under CO₂ atmosphere should not be simply attributed to the catalytic wave for CO₂ reduction. Reduction of protons or the Cu complex can also result in the formation of the wave. Electrolysis should be carried out to verify that the wave results from the reduction of CO₂.”

We thank the referee for the suggestion. We have performed bulk electrolysis of CuPP in CO₂-saturated DMF at -1.89 V (vs SCE) using a carbon rod working electrode (see graph below). We observed CO as the main gas product and only negligible amount of H₂ is generated. In a control experiment without CuPP, there is almost no CO or H₂ produced. Thus the catalytic wave in the CV is likely due to CO₂ reduction. We have added these results to the SI (Supplementary Fig. 19). We have revised the “Electrochemical analysis of CuPP” section and added the following sentences:

“Controlled potential coulometry experiments, performed at -1.89 V (vs SCE) in CO₂-saturated DMF, show that CO is the main gas product with a faradaic yield of 17% and only negligible amount of H₂ generated (Supplementary Fig. 20). In a control experiment without CuPP, there is little CO or H₂ produced. These data confirm that the observed current enhancement of CuPP corresponds to CO₂ reduction not proton reduction.”

Supplementary Figure 20. Bulk electrolysis time course for the amount of CO and H₂. Condition: with or without CuPP (10 µM) under CO₂-saturated DMF containing 0.1M TBAPF₆ at -1.89 V (vs SCE) using a carbon rod working electrode.

(6) “In Fig. 7, CuPP(6-) is proposed as the active catalyst. Can the authors speculate on the nature of this species? What is the oxidation state of Cu? What is the nature of the PP ligand? Is it simply a radical anion, are the carbonyl groups still intact?”

We thank the referee for the question. We have performed additional experiments to come up with a more complete picture of mechanism for CO₂ reduction. We have proposed a new mechanistic scheme, and revised the “Mechanistic studies” section, and added the following paragraph:

“CO generation was observed even at [BIH] as low as 0.1 mM. UV-vis spectra show no generation of additional absorption band in the visible region during CO₂ reduction (Fig. 8a), indicating CuPP is the active PS at this condition. Since [L₂Cu^{II}]⁻ (L = PP²⁻) has a much higher reduction potential (-1.05 V) than that of Fe(I)TDHPP (E_{1/2} = -1.55 V), CO₂ reduction has to proceed through oxidative quenching of L₂Cu^{II}* (-1.61 V) (Supplementary Table 4) in the photochemical step (Fig. 7). At high [BIH] (100 mM), a yellow species with absorption band at 400 nm is generated under visible light irradiation (Fig. 8b). When the reaction mixture is exposed to the air, the solution returns to purple color and over 80% of the original CuPP is recovered based on UV-vis spectra (Supplementary Fig. 34). Electrochemical studies in the presence of CO₂ (Fig. 5b) suggest that CuPP could undergo four steps of reduction and complexation of CO₂ to generate a [(CO₂)₂L₂Cu^{II}]⁴⁻ species (Fig. 7). In fact, PP has been shown to proceed a 2-electron reduction to generate a species at 398 nm during photocatalytic CO₂ reduction³⁴. Thus, we conclude that the intermediate with maximum absorption at 400 nm is a [(CO₂)₂L₂Cu^{II}]⁴⁻ species. [(CO₂)₂L₂Cu^I]⁵⁻ (-1.75V), generated by further reduction of [(CO₂)₂L₂Cu^{II}]⁴⁻ through reductive quenching, is corresponding to electron transfer to the Fe catalyst (Fig. 7).”

Fig. 7 Proposed reaction scheme for photocatalytic CO₂ reduction

Reviewer #3

We would like to thank Reviewer 3 for his/her comments and suggestions. Each point is addressed below.

“This manuscript describes the photocatalytic conversion of CO₂ to CO in a multi component system containing a Cu-based photosensitizer and a Fe porphyrin catalyst in DMF. Thus, the authors present a system which is based only on earth-abundant elements, with very good efficiency and selectivity for photocatalytic reduction of CO₂ to CO. Even though the system performs in organic solvent (and not in water) and therefore the high selectivity over proton reduction is expected, the notable aspect of the current study is the very good efficiency of the system, which is among the best reported to date for such systems. The other notable aspect of the study is the Cu-based photosensitizer.”

“This is a very nice study that includes a substantial amount of data, including photochemical, and electrochemical analyses. The work appears to be well done. The paper is well written, the experiments have been well performed with a range of techniques being employed. The mercury poisoning experiments are very convincing, together with all the other control experiments and the isotopic labeling experiments. The conclusions are well supported by the data presented. The results of this work are of interest in the AP community and to the larger research community as they do open an effective approach for the rational design of efficient chromophores in energy conversion systems.”

“I recommend publication after the following points are addressed.”

Thank you very much for your insightful reading of our manuscript and for your helpful comments. We have taken the referees' advice into account and performed a significant amount of further experiments. We have rewritten the manuscript to have a deeper discussion of the photocatalytic systems.

(1) “Better supporting the assignment of the UV-vis spectrum features of the CuPP photosensitizer (PS) based on literature or TD-DFT calculations. Did the authors do or do they plan to do TD-DFT calculations for this compound?”

Because the major absorption bands of CuPP show very similar feature as compared to the di-anion form of PP in the visible region (*J. Phys. Org. Chem.* **2000**, *13* (3), 141-150), the absorptions in the visible region for CuPP are assigned to the charge transfer bands on the PP moieties. Although the incorporation of Cu(II) does enhance this charge transfer process by showing a much higher molar extinction coefficient, these transitions are likely identical to PP. We have added a paragraph to the “Preparation and characterization of CuPP” section to describe the photophysical properties of CuPP and PP (please see our reply to question 2).

Due to the facility and time limitation, we are not able to perform TD-DFT calculations for CuPP. However, it would be very nice for our future study to use TD-DFT calculation to study how incorporating a Cu²⁺ center in tuning the HOMO and LOMO levels of the compound.

(2) “A more in-depth study of the photoluminescence properties of the CuPP would also be of interest. Comparisons of emission spectra of PP and CuPP obtained at different excitation wavelengths, in solution and in solid state, at room temperature and at low temperature would bring important information on the photophysical properties of the CuPP. I understand that it may be considered beyond the scope of this paper, but as a new PS is reported, information on its photoluminescence properties is of interest. In relation to this, what type of excited state do the authors propose for the CuPP? What are their thoughts about the geometry of the CuPP in the excited state?”

We thank the referee for the suggestion. We have included additional data to characterize the photophysical properties of CuPP and PP. We have added the following figures and table to the SI (Fig. 3b-c, Supplementary Fig. 11-14 and 35-36, and Supplementary Table 2). We have added the following paragraph to the “Preparation and characterization of CuPP” section to describe the photophysical properties of CuPP and PP:

“Because the UV-vis spectrum of CuPP exhibits identical feature as compared to the di-anion form of PP in the visible region⁵⁰, the absorption band at 566 nm is assigned to the charge transfer band on the PP ligands. The incorporation of Cu(II) enhances this charge transfer process by showing a much higher molar extinction coefficient (Supplementary Fig. 10). The photoluminescence quantum yield for CuPP (8.2×10^{-3}) is lower than that of PP (2.7×10^{-2}) (Supplementary Table 2). Instead, Romani and co-workers observed an increase of the fluorescence intensity when adding Al(III) to PP⁵¹. The distinct luminescent behavior of CuPP and Al-PP is probably due to the reason that the nature of emissions are based on different excited states. Excitation-emission spectra (Fig. 3b-c) show that the emission of CuPP is most intense when excited at 375 nm, which is similar to the charge transfer regions reported for Cu complexes^{52,53}. These data suggest that the emission of CuPP comes from a Cu(I) excited state generated from a LMCT process. The excitation spectrum of PP is comparable with the absorption spectrum (Supplementary Fig. 11a), which is in agreement with the literature⁵⁴. In contrast, the distinct absorption and excitation spectra of CuPP (Supplementary Fig. 11b) suggest a significantly structural change in its excited state, which is consistent with the observation that Cu(I) tends to adopt a tetrahedral geometry⁵⁵.”

Supplementary Table 2 Photophysical parameters of PP and CuPP at 298 K in DMF solution

Sample	Medium	Absorbance		Fluorescence				
		$\lambda_{\max}(\text{nm})$	$\epsilon_{\lambda_{\max}}(\text{M}^{-1} \text{cm}^{-1})$	$\lambda_{\max}(\text{nm})$	$\Phi_{\text{F}}^{\text{a}}$	τ (ns)	$\kappa_{\text{F}}/10^7$ (s ⁻¹)	$\kappa_{\text{NF}}/10^9$ (s ⁻¹)
PP	DMF	478	9940	582 ^b	2.7×10^{-2}	1.1	2.5	0.88
CuPP	DMF	566	38530	693 ^c	8.2×10^{-3}	0.98	0.84	1.01

^aMeasured under N₂, by absolute method using an integrating sphere, error 1-20 %. ^b $\lambda_{\text{exc}} = 484$ nm; ^c $\lambda_{\text{exc}} = 540$ nm.

Fig. 3. Excitation and emission spectra of PP (b) and CuPP (c) at different wavelengths at 298 K in DMF.

Supplementary Figure 13. Normalized emission spectra of PP (a, λ_{exc} :484 nm) and CuPP (b, λ_{exc} :540 nm) at 298 K and 77K in DMF.

Supplementary Figure 14. Solid state emission spectra of PP (a) and CuPP (b) with excitation wavelength of 350 nm at 298 K.

Supplementary Figure 35. Emission decay of PP (50 μM) in DMF at 298K (a) or 77K (b).

Supplementary Figure 36. Emission decay of CuPP (25 μM) in DMF at 298K (a) and 77K (b).

(3) “With respect to the electrochemical data, the reduction events in the CV of CuPP are difficult to analyse. How did the authors establish how many electrons each reduction is? Were also square-wave voltammetry experiments done for CuPP and PP? In addition, as CuPP is a new compound, showing the oxidation part of the electrochemical analysis of the CuPP vs PP (at least in the Supplementary Information) would be of interest too.”

We thank the referee for the suggestion. We have performed square-wave voltammetry experiments for both CuPP and PP to identify the number of electrons for each reduction (please see following figure) and have conducted the oxidation scans of CuPP and PP (Supplementary Fig. 19).

We have included the SWV results in our revised “Electrochemical analysis of CuPP” section as follows:

“..Square wave voltammetry (SWV) reveals that there are four successive reductions, which correspond to 2, 1, 1, and 2 electron processes (Supplementary Fig. 17)... The integrals of SWV show a total of four electron reduction at -1.13V prior to the catalytic current (Supplementary Fig. 17). The different electrochemical profiles observed for CuPP in the presence of CO₂ and N₂ indicate that the four quinone moieties on the PP ligands undergo four electron reductions and four complexations of CO₂ to generate carbonates. In fact, a similar observation for quinones have been studied in the literature⁵⁶...”

Supplementary Figure 17. SWV of 1 mM CuPP under N₂ (black solid) or CO₂ (red solid) in DMF containing 0.1 M TBAPF₆ at scan rate 0.1 V/s; dash lines show integrals for reduction waves. Inset: magnification of the SWV under N₂.

Supplementary Figure 18. SWV of 2 mM PP under N₂ (a) or CO₂ (b) in DMF containing 0.1M TBAPF₆ at 0.1 V/s scan rate.

Supplementary Figure 19. Oxidative part of CVs of 1 mM PP (a) and 1 mM CuPP (b) in DMF containing 0.1M TBAPF₆ at 0.1 V/s scan rate.

(4) “It will be interesting to see the profile of the TOF (at least in Supplementary Information) for the experiments shown in Fig. 4.”

We thank the referee for the suggestion. We have added Table 1 to the manuscript to include the profile of TOF:

Table 1 Control and other photocatalytic CO₂ reduction experiments to characterize the performance of the BIH/CuPP/FeTDHPP system in CO₂-saturated DMF solution

Entry	PS	[FeTDHPP] (μM)	[BIH] (mM)	Irradiation time (h)	CO (μmol)	H ₂ (μmol)	TON _{CO}	TOF _{CO} ^{max} (h ⁻¹)	TON _{H₂}	Sel _{CO} (%)
1	PP	0.2	10	7	0.088	0	88	45	0	100
2	CuPP	0.2	10	7	4.779	0.27	4779	1356	270	95
3	CuPP	0.2	100	23	16.109	0.843	16109	7650	843	95
4	CuPP	0	100	1	2.2	0	4.4 ^a	4.3	0	100

^aCalculated by the equation $\text{TON}_{\text{CO}} = n(\text{CO})/n(\text{CuPP})$

(5) “For the experiment shown in Fig. 4b, there is an induction time of almost 1 h, which is attributed to the low concentration of BIH. Can the decomposition of the CuPP be ruled out during this long induction time?”

We thank the referee for the question. We have performed experiments monitoring the reaction systems in real-time using UV-vis spectroscopy. These spectra were taken from the exact reaction conditions as Fig. 5a-b (previous Fig. 4a and 4b) to reflect the spectral changes on the course of photolysis (see following figure). The reduced CuPP species (at 400 nm), which could be oxidized back to CuPP (discussed in the revised manuscript in the “Mechanistic studies” section), starts to appear at around 1 hour. This species disappears at ~3.5 h, along with the cessation of CO production. This data suggests that the reduced CuPP is the active photosensitizer for CO₂ reduction. Thus, the decomposition of CuPP in the first 1h can be ruled out.

Supplementary Figure 29. UV-Vis absorption spectra of systems containing: (a) 0.1 mM CuPP, and 100 mM BIH; (b) 0.1 mM CuPP, 0.2 μM FeTDHPP and 10 mM BIH upon irradiation with white LED light in a quartz cuvette with 2 mm path length (dilution factor of 5).

(6) “Did the authors perform a control experiment with using Cu salt + PP instead of CuPP?”

We thank the referee for the question. We have performed the suggested control experiments using an *in situ* generated CuPP. The activity of this *in situ* generated photosensitizer is similar to the isolated CuPP (see following figures). In fact, we could observe the generation of CuPP in a solution mixture containing BIH, Cu^{2+} , and PP by UV-vis spectroscopy. We have added the following sentences to discuss the results in our revised “Photocatalytic CO_2 reduction” section:

“The CuPP PS can also be generated *in situ* with addition of 1:2 ratio of Cu^{2+} and PP. The amount of CO produced with the *in situ* generated PS is ~25% lower than that of the system using isolated CuPP (Supplementary Fig. 21a). UV-vis spectra confirm that CuPP can be generated in minutes when mixing Cu^{2+} and PP in the presence of BIH (Supplementary Fig. 21b).”

Supplementary Figure 21. (a) The amount of CO produced with the *in situ* generated CuPP (0.1 mM Cu^{2+} and 0.2 mM PP) (black) or with isolated CuPP (red) as the photosensitizers in CO_2 -saturated DMF solution containing 1.0 μM FeTDHPP and 10 mM BIH; (b) UV-vis spectra changes for a solution containing 0.1 mM Cu^{2+} , 0.2 mM PP, and 10 mM BIH in DMF.

(7) “How was the concentration of CuPP chosen? Did the authors perform experiments at different CuPP concentrations?”

We thank the referee for the questions. The concentration of CuPP was chosen to achieve optimal activity for CO production (see following figures). We have carried out additional experiments and added the following paragraph to the “Photocatalytic CO_2 reduction” section to describe the effects of concentration of photosensitizer and catalyst in catalysis:

“The activity of the photochemical systems were optimized further by varying the [CuPP] and [FeTDHPP]. At fixed concentrations of BIH (100 mM) and FeTDHPP (2 μM), increasing the CuPP concentration increases the overall rate of CO production (Supplementary Fig. 23). However, the rate of CO generation does not increase above 0.1 mM concentration of CuPP. When the CuPP concentration is fixed at 0.1 mM and [FeTDHPP] is varied (Supplementary Fig. 24), the catalyst is most active (on a TON basis) at low concentrations. The total amount of CO evolved increases up to 40 μM FeTDHPP, suggesting that the activity of the system becomes limited by electron transfer to the catalyst.”

Supplementary Figure 23. CO generation in CO₂-saturated DMF solutions containing 2 μM FeTDHPP and 100 mM BIH at different CuPP concentrations.

(8) “Did the authors perform any quantum yield measurements for this system?”

We thank the referee for the question. We have performed the measurement of quantum yield under 450nm monochromatic LED light. The initial (1 hour) quantum yield of CO₂ to CO conversion was determined to $6.0 \pm 0.6\%$. Related experimental details have been added to the method section. We have added the following sentence to the “Photocatalytic CO₂ reduction” section:

“For a system containing 2.0 μM FeTDHPP/100 mM BIH/0.1 mM CuPP, the initial 1 h quantum yield of CO₂ to CO conversion at 450 nm was determined to $6.0 \pm 0.6\%$.”

(9) “It is stated ‘Cu-O distances, electronic charge balancing, and paramagnetic proton resonances in the 1H NMR spectra (Supplementary Fig. 2) are all consistent with the assignment of a Cu(II) center.’ However no values and no references are given for the Cu-O distances. The distances Cu-O in CuPP are reported only in SI. It would be of interest to be given in the manuscript as well. In addition, a CCDC search reveals that there are several reports of the solid state structures of Cu catecholates, 1,2-naphthoquinones (e.g., CEJBUP, BAKPUY). Comparison in term of bond lengths with these structures would be of interest as well.”

We thank the referee for the suggestion. We have added the following table to the SI to compare the Cu-O bond lengths of related Cu catecholate compounds:

Supplementary Table 1 Cu-O bond lengths (Å) of Cu catecholate complexes

Cu-Complex	Cu-O1	Cu-O2	Reference
CuPP	1.9268(17) ^a	1.9168(18) ^a	This work
bis(catecholato)Cu	1.9165(4) ^a	1.9303(3) ^a	2
Cu(py) ₂ (Lw) ₂	2.454(2) ^b	1.945(2) ^c	3
Cu(Lw) ₂ (H ₂ O) ₂	2.336(2) ^b	1.954(3) ^a	4
Cu(L1) ₂ (EtOH) ₂	2.225(2) ^b	1.9301(17) ^a	5
Cu(Lap) ₂ (DMF) ₂	2.301(1) ^b	1.914(1) ^a	6
CuL ₂ py ₂	2.415(4) ^b	1.948(2) ^c	7

^aphenoxy coordination; ^bquinonic carbonyl coordination; ^cenolic coordination;

We have added related references and revised this sentence in the manuscript text as follows:
“The Cu-oxygen distances (Cu-O1 = 1.927(2) and Cu-O2 = 1.917(2) Å) are similar to the Cu-O bonds reported for Cu(II) catecholates compounds in the literature (Supplementary Table 1). This evidence along with the electronic charge balancing, and paramagnetic proton resonances in the ^1H NMR spectra (Supplementary Fig.3-4) are all consistent with the assignment of a Cu(II) center in CuPP.”

(10) “The authors state in the Conclusion: ‘For example, the CuPP/FeTDHPP system achieves over 16100 turnovers of CO with a maximum TOF of 7650 h^{-1} , which is 2 orders of magnitude more than a reported $\text{Ir}(\text{ppy})_3/\text{FeTDHPP}$ system (TON = 140 in 55 h)³¹, and shows an almost 4-fold increase of a $[\text{Ru}(\text{bpy})_3]^{2+}/\text{CoTPPS}$ system (TON = 4000, $\text{TOF}_{\text{max}} = 2400\text{ h}^{-1}$)⁵¹.’ The comparison is very pertinent with respect to the $\text{Ir}(\text{ppy})_3/\text{FeTDHPP}$ system which is a system in organic solvent. However a distinction should be made with respect to the $[\text{Ru}(\text{bpy})_3]^{2+}/\text{CoTPPS}$ system, which is a system performing in water.”

We agree with the referee that this comparison was confusing. We have revised the manuscript text and deleted the statement regarding to the comparison with the $[\text{Ru}(\text{bpy})_3]^{2+}/\text{CoTPPS}$ system.

(11) “In the Supplementary table 2 and Supplementary table 3, it will be of interest to add the solvent in which the system performs.”

We thank the referee for the suggestion. We have added the solvents used for all systems listed in supplementary table 5-8 (previous Supplementary Table 1-4).

(12) “Supplementary Figure 1. ESI-MS spectra of $\text{Na}_2\text{Cu}(\text{PP})_2$ in CH_3OH (negative ion mode). The most intense signals was at m/z ($[\text{M}-\text{Na}^+]$) = 594.22 (calcd: 593.96).’ There is an important difference between the experimental and the calculated m/z . In addition, the comparison between the experimental isotopic pattern and the calculated isotopic pattern should also be presented. Did the authors also perform the MS for the $\text{Cu}(\text{PP})_2(\text{TBA})_2$?”

We thank the referee for the comment. We have performed high-resolution mass spectra for $\text{Na}_2\text{Cu}(\text{PP})_2$ (see following figures). The measured and calculated m/z are both 593.96 and the isotopic patterns are identical. We have revised the SI to include these results.

Supplementary Figure 1. HRMS spectra of $\text{Na}_2\text{Cu}(\text{PP})_2$ (a) in CH_3OH (negative ion mode).

We have also performed the high-resolution mass spectra for $\text{Cu}(\text{PP})_2(\text{TBA})_2$. We have revised the SI to include these results. The overall spectrum and fragment signals are shown as follows:

Supplementary Figure 1. HRMS spectra of $(\text{TBA})_2(\text{CuPP})_2$ (b) in CH_3OH (negative ion mode).

Supplementary Figure 2. HRMS spectra (enlargement of Supplementary Figure 1b) of $(TBA)_2Cu(PP)_2$ in CH_3OH (negative ion mode). The most intense signals was at m/z $[(PP)^2 + H^+]^- = 255.03000$ (calcd: 25.03000) (a); A peak at $m/z = 571.98047$ (calcd: 571.98102) corresponds to a fragment of $[(TBA)_2Cu(PP)_2 - 2TBA^+ + H^+]^-$ (b); A peak at $m/z = 813.25719$ (calcd: 813.25797) corresponds to of $[(TBA)_2Cu(PP)_2 - TBA^+]^-$ (c).

(13) “Supplementary Figure 11. UV-Vis absorption spectra of systems containing 5.0 μM CuPP, 0.2 μM FeTDHPP, and 10 mM BIH upon irradiation with white LED light.’ To show a zoom on the region 450 – 650 nm would be of interest too.”

We thank the referee for the suggestion. In the previous experiments, we monitored the UV-vis spectral changes by conducting photocatalytic CO_2 reduction in a cuvette. We have revised the figure by taking the spectra at the exact reaction conditions as shown in Fig. 5c to measure the actual spectral changes on the course of photolysis (see following figure). The updated figure is presented as follows:

Fig. 8 UV-vis absorption spectra of systems containing (b) 0.1 mM CuPP, 0.2 μM FeTDHPP and 100 mM BIH upon irradiation with white LED light. Solutions for (b) at different times were transferred and diluted 5 times with DMF to a quartz cuvette (2 mm path length) under N_2 .

(14) *“Have CH₄ also been quantified in this system?”*

Unfortunately, CH₄ has not been detected in our system using a FID detector in gas chromatography.

(15) *“Could the CuPP and the catalyst also have favourable electrostatic interactions? Could this play a role in the enhanced performance?”*

In Supplementary Figure 30, we do not see any change from the UV-vis experiments when adding FeTDHPP catalyst to a CuPP solution. Thus, we do not think there is any electrostatic interaction between them.

Minor points:

(16) *“As a suggestion, the paper would benefit by moving the Fig. 5 (Illustrative scheme of the photocatalytic CO₂ reduction system investigated in this study) at the beginning of the manuscript before the Results, where the authors present the system under discussion in this work.”*

Thank you for the suggestion. We agree with the reviewer to move Fig.5 to the front the results. Now it is shown in Fig. 1.

(17) *“It is written ‘The Yield 21.8%’. Reporting the overall yield of this type of synthesis with such precision is difficult. The value should be 22%.”*

This error has been corrected.

(18) *“In Supplementary table 1, in the footnote caption, the description of entries 6 and 7 are missing.”*

This error has been corrected for previous Table 1 (now revised Table 4)

(19) *“The excitation wavelength should be mentioned in the Supplementary Figure 13 and the Supplementary Figure 14.”*

This error has been corrected. We have added the excitation wavelength to the Supplementary Figure 31-32 (previous Supplementary Figure 13-14).

(20) *“In the manuscript as section title, it is written Discussion instead of Conclusion.”*

Thank you for pointing this imprecision out. Because Nature Communications does not have a separated section for “Conclusion”, we have changed the section title “Result” into “Result and Discussion” and deleted the section title “Discussion” for the last paragraph.

REVIEWER COMMENTS

Reviewer #1 (Remarks to the Author):

I thank the authors for their careful replies. I really appreciate the amount of new experiments and the clarifications they made to make this scientific contribution more incisive. Their replies are mostly satisfactory and the manuscript has been improved greatly. Nevertheless, there is still an important issue that has to be clarified before publication. This is about the nature of photophysical properties of the new Cu(II) complex of purpurin (CuPP) that is used as photosensitizer and might work also as catalyst in photocatalytic CO₂ reduction.

The absorption spectrum of purpurin is different from the coordinated metal complexes thereof, because the purpurin in the metal complex is negatively charged. In fact, as we can see the absorption spectra of Cu(II)purpurin (CuPP) and Al(III)purpurin (in the cited article) have very similar profiles. The different values in photoluminescence quantum yield (PLQY) of its complexes made by coordination of Cu(II) and Al(III) is certainly due the different nature of metal ions, being the first one paramagnetic and the latter diamagnetic, as also the authors point it out. Nevertheless, while the emission bands in Al(III) purpurin complex are the mirror image (with a short Stokes shift) of the absorption bands, typical of fluorescence processes, the emission of CuPP is red-shifted and it is structureless. The authors explain the bathochromic shift of the emission of the CuPP in respect to a "different nature of the excited state".

Although I am in agreement with the authors that a different process might be responsible of the measured emission, I am not satisfied with the explanation the authors give, because:

a) The authors compare the Cu(II) complex CuPP with Cu(I) complexes in the literature (Angew. Chem. 2013, 125, 437-441), although Cu(I) complexes have very different behaviours. Moreover, they cite an old paper focusing on Raman spectroscopy (J. Am. Chem. Soc. 1974, 96, 6868-6873). The authors should cite appropriate literature, that is referring to photosensitizers based on Cu(II) complexes. Some examples could be:

- A. Hossain et al. Angew Chem Int. Ed. 2018, 57, 8288;
- Y. Li et al. J. Am. Chem. Soc. 2018, 140, 15850;

b) If a formally Cu(I) species is formed during excitation in the electronic transition, responsible of the population of a LMCT state with geometrical distortion, proposed by the authors, the population of this excited state is normally followed by a release of a ligand. (see for example the literature cited in (a)). What is the suggestion of the authors?

c) If CuPP upon excitation populates this LMCT and becomes formally a Cu(I) species. Then it should be true also for the proposed mechanistic cycle at low BIH concentration, but the authors report only Cu(II) species in that cycle (Fig.7). A Cu(I) species appears only in the proposed mechanism at high BIH concentration. However, in that cycle the photosensitizer species is not the original CuPP, but a reduced CuPP that is coordinating four CO₂ molecules. Therefore the absorption and emissive properties of this new compound should be different from the simple CuPP. Can the authors comment on this?

Further, I know that the homogeneous photocatalytic reduction of CO₂ is a hot topic and it is difficult to cite all the new articles. Nevertheless, last year other important contributions about CO₂ photocatalytic reduction containing non-noble metals or organic photosensitizers have appeared and should be cited. For example:

- Dalton Trans., 2020, 49, 4230-4243
- Chem. Eur. J., 2020, 26, 9929-9937
- Chem. Eur. J, 2020, 26, 16373
- J. Am. Chem. Soc. 2020, 142, 705
- Chem. Commun. 2020, 56, 12170.

Reviewer #2 (Remarks to the Author):

In the opinion of this reviewer, the authors have made adequate revisions to their manuscript and it is now suitable for publication in nature Communications.

Reviewer #3 (Remarks to the Author):

This manuscript describes the photocatalytic conversion of CO₂ to CO in a multi component system containing a Cu-based photosensitizer and a Fe porphyrin catalyst in DMF. Thus, the authors present a system which is based only on earth-abundant elements, with very good efficiency and selectivity for photocatalytic reduction of CO₂ to CO. The notable aspects of the current study are the very good efficiency of the system (which is among the best reported to date for such systems) and the Cu-based photosensitizer.

This is a very nice study that includes a substantial amount of data, including photochemical, and electrochemical analyses. The paper is well written, the experiments have been well performed with a range of techniques being employed. The mercury poisoning experiments are very convincing, together with all the other control experiments and the isotopic labeling experiments. The conclusions are well supported by the data presented. The results of this work are of interest in the AP community and to the larger research community as they do open an effective approach for the rational design of efficient chromophores in energy conversion systems.

I am pleased to recommend publication of this manuscript.

A minor point:

The following part added on the quantum yield of the system needs to be reviewed as there are minor inconsistencies (e.g., $n(\text{CO})$ and I are described, but there are not written as such in the formula of quantum yield).

Reviewer #1:

We would like to thank Reviewer 1 for his/her helpful comments and critical reading of our manuscript, which enabled us to improve the clarity and quality of our work. Each point is addressed below.

“I thank the authors for their careful replies. I really appreciate the amount of new experiments and the clarifications they made to make this scientific contribution more incisive. Their replies are mostly satisfactory and the manuscript has been improved greatly. Nevertheless, there is still an important issue that has to be clarified before publication. This is about the nature of photophysical properties of the new Cu(II) complex of purpurin (CuPP) that is used as photosensitizer and might work also as catalyst in photocatalytic CO₂ reduction.”

“The absorption spectrum of purpurin is different from the coordinated metal complexes thereof, because the purpurin in the metal complex is negatively charged. In fact, as we can see the absorption spectra of Cu(II)purpurin (CuPP) and Al(III)purpurin (in the cited article) have very similar profiles. The different values in photoluminescence quantum yield (PLQY) of its complexes made by coordination of Cu(II) and Al(III) is certainly due the different nature of metal ions, being the first one paramagnetic and the latter diamagnetic, as also the authors point it out. Nevertheless, while the emission bands in Al(III) purpurin complex are the mirror image (with a short Stokes shift) of the absorption bands, typical of fluorescence processes, the emission of CuPP is red-shifted and it is structureless. The authors explain the bathochromic shift of the emission of the CuPP in respect to a “different nature of the excited state”. Although I am in agreement with the authors that a different process might be responsible of the measured emission, I am not satisfied with the explanation the authors give, because:”

*“a) The authors compare the Cu(II) complex CuPP with Cu(I) complexes in the literature (Angew. Chem. 2013, 125, 437-441), although Cu(I) complexes have very different behaviours. Moreover, they cite an old paper focusing on Raman spectroscopy (J. Am. Chem. Soc. 1974, 96, 6868-6873). The authors should cite appropriate literature, The authors should cite appropriate literature, that is referring to photosensitizers based on Cu(II) complexes. Some examples could be:
- A. Hossain et al. Angew Chem Int. Ed. 2018, 57, 8288;
- Y. Li et al. J. Am. Chem. Soc. 2018, 140,15850;”*

We thank the referee for pointing this out. We agree with the referee that the cited paper (J. Am. Chem. Soc. 1974, 96, 6868-6873) is not directly related to CuPP. We have added the suggested papers to the revised manuscript and replaced this JACS paper with a more appropriate one (Chem. Rev. 1998, 98, 1201-1220).

We have revised the “Preparation and characterization of CuPP” section:

“...Instead, Romani and co-workers observed an increase of the fluorescence intensity when adding Al(III) to PP⁵⁵. Short Stokes shifts of emissions were observed for both Al-PP and PP, and their excitation spectra are comparable with the absorption spectra (Supplementary Fig. 11a)^{55,56}, suggesting that these compounds undergo relatively low structural reorganization between the

ground and excited states. In contrast, CuPP displays structureless emission at 693 nm with a large bathochromic shift. The distinct absorption and emission spectra of CuPP (Supplementary Fig. 12b) indicates a significantly structural change in its excited state. Since Cu(I) complexes tend to adopt distorted geometries compared to their Cu(II) analogues^{40,57-60}, the nature of the emission of CuPP is likely from a Cu(I) excited state generated from a LMCT process. Excitation-emission spectra (Fig. 3b-c) show that the emission of CuPP is most intense when excited at 375 nm. The excited-state lifetimes are on the nanosecond time-scale for both CuPP ($\tau = 0.98$ ns) and PP ($\tau = 1.1$ ns)^{35,54}.”

“b) If a formally Cu(I) species is formed during excitation in the electronic transition, responsible of the population of a LMCT state with geometrical distortion, proposed by the authors, the population of this excited state is normally followed by a release of a ligand. (see for example the literature cited in (a). What is the suggestion of the authors?”

We thank the referee for the question. We believe CuPP is stable in its excited state based on UV-vis, electrochemical, ¹H NMR, and photocatalytic study. Firstly, we do not observe any dissociated PP ligand in the ¹H NMR spectra when irradiating a BIH/CuPP solution for 2 hours (please see following figure). Secondly, multiple CV scans (Supplementary Fig. 16) show that the reduction waves of CuPP are reproducible during these scans, suggesting the resulting Cu(I) species is intact. Furthermore, the UV-vis spectral profile of CuPP remains identical during CO₂ reduction over a few hours at low BIH concentration (Figure 8a), which supports that CuPP does not release a PP ligand in the catalytic cycle. The different observation between CuPP and the Cu complexes in the cited papers (*Angew Chem Int. Ed.* 2018, 57, 8288; *J. Am. Chem. Soc.* 2018, 140,15850) is probably due to that the monodentate ligands in their study are easier to dissociate during photo-oxidation than PP.

Figure ¹H NMR spectra of 50 mM BIH (a), 10 mM PP (b), and a mixture of CuPP(5 mM)/BIH (50 mM) after irradiation with white LED light for 2 h (c) in *d*₆-DMSO under N₂.

“c) If CuPP upon excitation populates this LMCT and becomes formally a Cu(I) species. Then it should be true also for the proposed mechanistic cycle at low BIH concentration, but the authors report only Cu(II) species in that cycle(Fig.7). A Cu(I) species appears only in the proposed mechanism at high BIH concentration. However, in that cycle the photosensitizer species is not the original CuPP, but a reduced CuPP that is coordinating four CO₂ molecules. Therefore the absorption and emissive properties of this new compound should be different from the simple CuPP. Can the authors comment on this?”

We thank the referee for the comment. We agree with the referee that a Cu(I) species is involved in the catalytic cycle at low BIH concentration. We have changed “L₂Cu^{II*}” to “(L₂)⁺Cu^{I*}” in the proposed reaction scheme (Fig. 7). Regarding to the mechanism at high BIH concentration, since we could not isolate the carbonate intermediates, we do not have direct evidence for a Cu(I) species. We proposed the Cu(I) intermediate based on square wave voltammetry (SWV) study. The integrals of SWV show a total of four electron reduction prior to the catalytic current (Supplementary Fig. 17). This suggests the CO₂ reduction occurs upon reduction of CuPP to its Cu(I) or even lower oxidation state.

“Further, I know that the homogeneous photocatalytic reduction of CO₂ is a hot topic and it is difficult to cite all the new articles. Nevertheless, last year other important contributions about CO₂ photocatalytic reduction containing non-noble metals or organic photosensitizers have appeared and should be cited. For example:

- Dalton Trans., 2020,49, 4230-4243*
- Chem. Eur. J., 2020, 26, 9929-9937*
- Chem. Eur. J, 2020, 26, 16373*
- J. Am. Chem. Soc. 2020, 142, 705*
- Chem. Commun. 2020, 56, 12170.”*

We would like to thank the referee for helping with the references. We have now cited these recent papers in the revised manuscript and SI.

Reviewer #2:

“In the opinion of this reviewer, the authors have made adequate revisions to their manuscript and it is now suitable for publication in nature Communications.”

Reviewer #3:

“This manuscript describes the photocatalytic conversion of CO₂ to CO in a multi component system containing a Cu-based photosensitizer and a Fe porphyrin catalyst in DMF. Thus, the authors present a system which is based only on earth-abundant elements, with very good efficiency and selectivity for photocatalytic reduction of CO₂ to CO. The notable aspects of the current study are the very good efficiency of the system (which is among the best reported to date for such systems) and the Cu-based photosensitizer.”

“This is a very nice study that includes a substantial amount of data, including photochemical, and electrochemical analyses. The paper is well written, the experiments have been well performed with a range of techniques being employed. The mercury poisoning experiments are very convincing, together with all the other control experiments and the isotopic labeling experiments. The conclusions are well supported by the data presented. The results of this work are of interest in the AP community and to the larger research community as they do open an effective approach for the rational design of efficient chromophores in energy conversion systems.”

“I am pleased to recommend publication of this manuscript.”

We would like to thank Reviewer 3 for his/her helpful comments and critical reading of our manuscript, which enabled us to improve the clarity and quality of our work.

“A minor point:

The following part added on the quantum yield of the system needs to be reviewed as there are minor inconsistencies (e.g., $n(\text{CO})$ and I are described, but there are not written as such in the formula of quantum yield).”

We regret the mistake and have revised the equations for calculating quantum yield and have clarified $n(\text{CO})$ and I in the experimental section.

REVIEWERS' COMMENTS

Reviewer #1 (Remarks to the Author):

I thank the authors for further explanations and corrections/additions to the manuscript.
I am pleased to recommend the manuscript at the present stage for publication.

Reviewer #1:

*I thank the authors for further explanations and corrections/additions to the manuscript.
I am pleased to recommend the manuscript at the present stage for publication.*

We would like to thank Reviewer 1 for his/her helpful comments and critical reading of our manuscript, which enabled us to improve the clarity and quality of our work.